# Fingerprinting Pre-trained Encoders under Arbitrary Downstream Fine-Tuning via Adversarial Shifting

**Tianlong Xu** [1] **Zixiong Wang** [1] **Lishuai Hou** [1] **Gaoyang Liu** [1] **Chen Wang** [1] **Xiaoyi Fan** [2]

## Abstract

In the pre-training-fine-tuning paradigm, pre-trained encoders have become high-value intellectual property (IP) due to their immense training costs, necessitating robust protection. Existing fingerprinting or watermarking methods typically rely on pre-defined samples and labels, or require intrusive modifications to the training process. However, downstream fine-tuning can significantly alter an encoder's representation and label space, thereby destroying the label consistency of existing methods and rendering them ineffective. Consequently, it is both challenging and urgent to provide a downstream-agnostic, black-box ownership verification mechanism for pre-trained encoders. To address this, we propose a downstream-agnostic, label-only fingerprinting method that leverages Adversarial Shifting to construct stable fingerprint clusters in the encoder's latent space. By exploiting the inherent output consistency of these clusters, our method remains effective regardless of the specific downstream task or label mapping. Extensive experiments demonstrate that our method maintains superior robustness and stealthiness across various downstream tasks and category scales, providing a practical and reliable IP protection scheme for high-value pre-trained encoders. The code is available at: https://github.com/SPHelixLab/EncoderFingerprint.

## 1. Introduction

Deep learning has witnessed a paradigm shift toward using pre-trained encoders as foundational backbones for diverse downstream tasks. This "pre-trained encoder + re-trained head" paradigm significantly reduces data dependency while enhancing generalization (Ren et al., 2025; Fang et al., 2025). As the core carrier of transferable knowledge, the encoder accounts for the majority of computational overhead and parameters, embodying substantial Intellectual Property (IP) value—a trend further accelerated by the rise of Encoder-as-a-Service (EaaS) (Qu et al., 2023). Consequently, these high-value encoders are prime targets for theft (Liu et al., 2022). Adversaries can illicitly obtain an encoder, integrate it with arbitrary heads, and deploy it as a black-box service. Under such settings, verifying ownership through prediction labels remains a formidable challenge, as downstream fine-tuning fundamentally alters output semantics (Zhou & Srikumar, 2022; Tětková et al., 2025).

In practical deployment, fine-tuning pre-trained encoders on heterogeneous downstream tasks induces substantial discrepancies in label spaces and representation semantics (Zhou et al., 2023b; Wu et al., 2022). This characteristic fundamentally invalidates existing fingerprinting methods (e.g., ADV-TRA (Xu et al., 2024), MFUE (Xu et al., 2026)) that rely on pre-defined fingerprint labels. These approaches necessitate output alignment between the suspect and victim models—a requirement rarely satisfied in realistic scenarios. Consequently, traditional label-dependent verification mechanisms become ineffective when applied to fine-tuned encoders. On the other hand, while recent works (e.g., SSL-WM (Lv et al., 2024a), MEA (Lv et al., 2024b), SSLGuard (Cong et al., 2022)) have attempted to embed watermarks into Self-Supervised Learning (SSL) encoders, they suffer from significant limitations: (1) *Intrusiveness*, as they require modifying training objectives (e.g., contrastive loss), potentially degrading performance; and (2) *Verification Dilemma*, as they typically depend on internal embeddings that are inaccessible in black-box, label-only API scenarios. Under the practical pre-training + fine-tuning paradigm, existing studies have yet to provide a downstream-agnostic black-box fingerprinting-based ownership verification mechanism for pre-trained encoders.

To bridge this gap, we analyze the propagation behavior of adversarial examples within deep networks (Liu et al., 2025), analyzing their manifestations across different layers (see

[1]Hubei Key Laboratory of Internet of Intelligence, School of EIC, Huazhong University of Science and Technology, Wuhan, China [2]Jiangxing Intelligence Technology Inc., Guizhou, China. Correspondence to: Gaoyang Liu <liugaoyang@hust.edu.cn>.

*Proceedings of the $43^{rd}$ International Conference on Machine Learning*, Seoul, South Korea. PMLR 306, 2026. Copyright 2026 by the author(s).

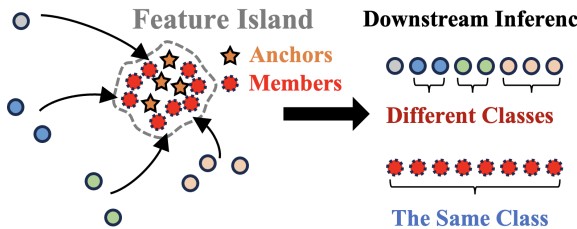

*Figure 1.* Illustration of Adversarial Shifting. Samples from other clusters are adversarially perturbed to migrate towards the target region (vicinity of anchors) to form members. This ensures that these members are consistently predicted as the same class, regardless of the specific downstream task.

Figure 11 in Appendix). We observe a key phenomenon: distributional deviations from normal samples remain negligible in shallow layers but amplify significantly toward the encoder's output. This explains why traditional output-dependent fingerprints fail—shallow features lack identifiable patterns (Zhou et al., 2023a; 2024). Leveraging this, we propose an endogenous fingerprint via adversarial shifting in the feature space. Specifically, we perturb a group of samples to aggregate within a target cluster, forming a cohesive "feature island" in the representation domain (as shown in Figure 1). Due to this high intra-group proximity, these perturbed samples (referred to as members) are highly likely to yield consistent predictions across arbitrary downstream heads, enabling task-agnostic ownership verification.

However, translating this insight into a robust and practical verification mechanism still entails two primary challenges: (1) Targeting in the Feature Space: Precise manifold positioning is difficult without downstream task knowledge (Cohen et al., 2020; Zhang et al., 2025). We propose an unsupervised anchor-guided Shifting strategy, selecting anchors from dense clusters to guide fingerprints into a distinguishable "feature island." (2) Task-Agnostic Black-box Verification: Unpredictable output spaces in black-box APIs render traditional label-matching ineffective. We introduce group fingerprinting, which leverages high feature-space cohesion rather than individual labels. Since these members exhibit near-identical representations, any classifier will yield consistent predictions. Ownership is thus verified via voting concentration. This approach successfully decouples verification from task priors, ensuring robustness even under label-only and heterogeneous output conditions.

To summarize, our contributions are as follows:

- We propose a universal encoder fingerprinting framework compatible with both supervised and self-supervised paradigms, providing scalable IP protection for diverse high-value pre-trained models.

- We introduce an adversarial shifting-based generation method that constructs endogenous fingerprints by migrat-

ing samples toward a same semantic cluster. To enable label-only verification, we design a group voting algorithm that exploits output consistency among clustered samples, eliminating the access of encoder internals.

- Extensive experiments confirm our fingerprints' superior uniqueness and robustness. Furthermore, we reveal that feature-space proximity effectively regulates adversarial transferability, offering new heuristic insights for adversarial research.

## 2. Preliminary

### 2.1. Model Fingerprinting

Model fingerprinting serves as a non-invasive approach for IP protection, verifying model ownership by extracting unique behavioral patterns without modifying model parameters (Cao et al., 2021; Peng et al., 2022). Formally, let $f : \mathcal{X} \to \mathcal{Y}$ be a classification model. A fingerprint set $\mathcal{D}_{fp} = \{(x_{fp}^{(i)}, y_{fp}^{(i)})\}_{i=1}^{L}$ typically consists of some special adversarial examples optimized to trigger specific predictions $y_{fp} \neq y_{true}$ on $f$:

$$x_{fp} = x + \delta, \quad \text{s.t.} \ f(x_{fp}) = y_{fp}. \quad (1)$$

Ownership is verified by querying a suspect model $f_{sus}$ and calculating a statistical metric, such as the following, to measure the alignment between the suspect model's predictions and the pre-defined labels:

$$\text{Metric} = \frac{1}{L} \sum_{(x_{fp}, y_{fp}) \in \mathcal{D}_{fp}} \mathbb{I}(f_{sus}(x_{fp}) = y_{fp}). \quad (2)$$

If Metric exceeds a threshold, $f_{sus}$ is identified as an infringing copy.

Extending fingerprinting to pre-trained encoders ($E$) is non-trivial because the final model is a composition $f = E \circ H$, where the downstream head $H$ is unknown. Existing methods fail in our threat model due to : (1) **Unknown Decision Boundaries:** Since $H$ is trained by the attacker on a private dataset $\mathcal{D}_{down}$, the decision boundaries are non-existent during fingerprint generation. (2) **Inaccessible Feature Space:** Under the black-box setting, the defender only observes the labels from $f(x)$, not the intermediate embedding $z = E(x)$. Consequently, robust encoder fingerprints must be anchored in the invariant feature space rather than the volatile label space.

### 2.2. Self-Supervised Learning

Self-supervised learning pre-trains encoders to learn generalized feature representations from unlabeled data. Taking Contrastive Learning (e.g., SimCLR) as a representative paradigm, the objective is to maximize the similarity between positive pairs (augmentations of the same instance)

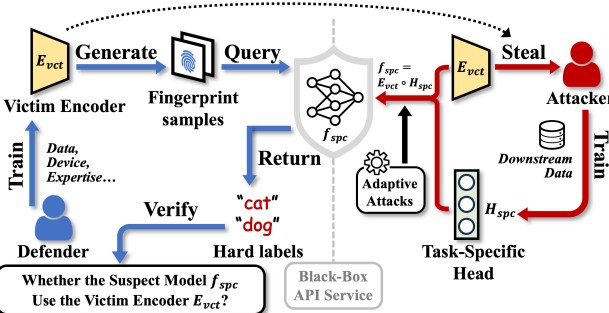

*Figure 2.* Overview of the threat model for pre-trained encoder ownership verification

in the latent space. Given an image $x$, the pipeline generates two stochastic views forming a positive pair. The encoder $f(\cdot)$ extracts features, which a projection head $g(\cdot)$ maps to a latent space to compute the contrastive loss:

$$\mathcal{L} = -\log \frac{\exp(\text{sim}(z_i, z_j)/\tau)}{\sum_{k=1}^{2N} \mathbb{I}_{[k \neq i]} \exp(\text{sim}(z_i, z_k)/\tau)}, \quad (3)$$

where $z$ denotes the feature embedding and $\tau$ is a temperature parameter. Essentially, minimizing this objective forces the encoder to pull representations of positive pairs closer together while pushing negative pairs apart, thereby learning a discriminative feature space invariant to augmentations.

Our study also incorporates several advanced SSL algorithms. For example, MoCo V2 employs a momentum encoder and a dynamic queue to maintain a large set of negative samples, decoupling the dictionary size from mini-batch size. SigLIP utilizes a sigmoid-based loss to optimize image-text contrastive learning, offering superior efficiency for large-scale visual pre-training.

## 3. Threat Model

We consider a realistic IP protection scenario involving a pre-trained encoder owner (the **Defender**) and an unauthorized user (the **Attacker**). Given only black-box, label-only access to a suspect model, the defender aims to verify whether the model has been fine-tuned from the protected pre-trained encoder. In this work, we focus on encoder fingerprinting as a non-invasive IP protection strategy, which verifies encoder ownership by exploiting unique and identifiable behaviors on carefully crafted fingerprint samples. Figure 2 illustrates this scenario, delineating the interactions and constraints between the two parties. Due to space limitations, related works are provided in Appendix A.

**Attacker.** The attacker steals the victim encoder $E_{vct}$ through unauthorized means and aims to exploit it for profit by building a downstream service.

- **Knowledge.** The attacker has full white-box access to the

stolen encoder $E_{vct}$. They utilize their own private dataset $\mathcal{D}_{down}$ to train a task-specific head $H_{spc}$ on top of $E_{vct}$, composing the infringing model $f_{spc} = E_{vct} \circ H_{spc}$.

- **Capabilities.** To evade ownership verification, the attacker may employ various strategies to erase potential fingerprints or watermarks while maintaining model utility. These strategies include model modifications such as fine-tuning and pruning, as detailed in the experimental setup (Section D).

- **Goal.** The attacker aims to remove or obfuscate the encoder's original identity markers, thereby precluding the defender from verifying the ownership of the deployed model $f_{spc}$.

**Defender.** The defender is the legitimate owner who has expended significant computational resources, data, and expertise to train a high-performance victim encoder $E_{vct}$ (e.g., via supervised or self-supervised learning).

- **Knowledge.** The defender has white-box access to $E_{vct}$ for generating fingerprints. Crucially, the defender operates in a downstream-agnostic manner: they have zero prior knowledge of the attacker's specific downstream tasks, the downstream head $H_{spc}$, or the distribution of the downstream dataset $\mathcal{D}_{down}$.

- **Capabilities.** During the verification phase, the defender is restricted to black-box, *label-only* access to the suspect model $f_{spc}$. The Defender can only query the model's API and observe the final predicted labels. Intermediate outputs, such as feature embeddings or soft labels (logits), are strictly inaccessible.

- **Goal.** The Defender aims to establish irrefutable evidence of ownership by detecting a persistent fingerprint that survives the attacker's task-specific adaptation and remains verifiable through the black-box API.

## 4. Methodology

### 4.1. Overview

The core objective of our approach is to construct an endogenous fingerprint within the *Victim Encoder* $E_{vct}$ that remains invariant to the downstream fine-tuning. As illustrated in Figure 3, our methodology consists of two primary phases: *Group Fingerprint Generation* and *Ownership Verification*. In the generation phase, we introduce **Adversarial Shifting** to migrate a group of samples into a cohesive "feature island" within the feature space of $E_{vct}$, guided by unsupervised *Anchors*. This process ensures that the optimized samples, referred to as *Members*, exhibit extreme proximity in the representation domain. Consequently, in the verification phase, we employ a **Group Voting** mechanism. By

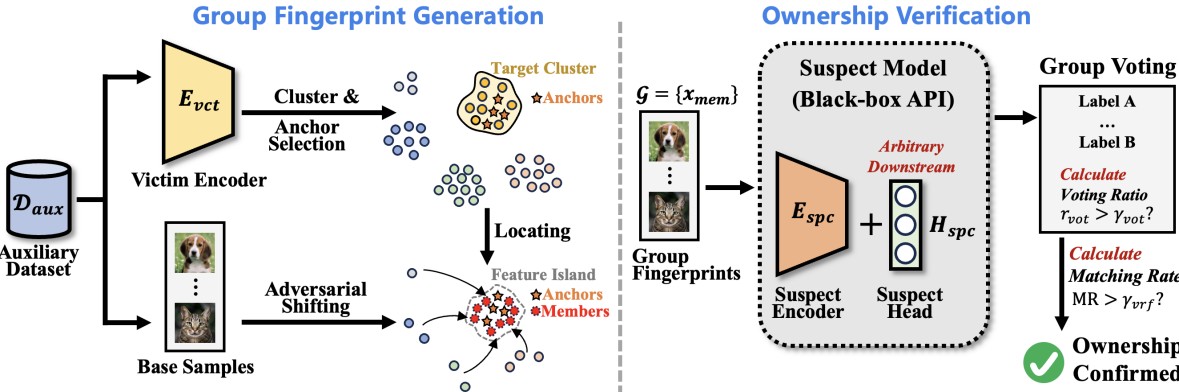

*Figure 3.* Framework of Feature Space-based Adversarial Fingerprinting. The process is divided into two phases: (1) **Fingerprint Generation**, where base samples are adversarially shifted in the encoder's feature space to form a tight "feature island" guided by anchors. (2) **Ownership Verification**, where the generated group fingerprints are queried against a black-box suspect model. Ownership is verified if the group's predicted labels show high consistency through a voting mechanism.

querying the black-box *Suspect Model* $f_{spc}$ with these members, we observe their output consistency. If the statistical agreement exceeds a voting threshold, the ownership of the encoder is confirmed, regardless of the downstream task's specific semantics.

### 4.2. Group Fingerprint Generation

The generation of robust fingerprints for an encoder requires identifying regions in the high-dimensional feature space that are likely to be preserved during downstream adaptation. We achieve this through three steps: feature space characterization, anchor selection, and adversarial shifting.

#### 4.2.1. FEATURE SPACE CHARACTERIZATION

Let $E_{vct} : \mathcal{X} \to \mathcal{Z}$ denote the victim encoder that maps input samples to a high-dimensional feature space $\mathcal{Z} \subset \mathbb{R}^d$. Given an auxiliary dataset $\mathcal{D}_{aux}$ (unlabeled or labeled) available to the defender, we first extracts the high-dimensional feature representations:

$$\mathcal{Z}_{aux} = \{\mathbf{z}_i \mid \mathbf{z}_i = E_{vct}(x_i), x_i \in \mathcal{D}_{aux}\}, \qquad (4)$$

where $\mathbf{z}_i \in \mathbb{R}^d$ denotes the embedding vector. Since the raw feature space is high-dimensional and complex, we employ Spectral Clustering to partition $\mathcal{Z}_{aux}$ into $K$ clusters $\mathcal{C} = \{C_1, C_2, \ldots, C_K\}$. Unlike distance-based methods (e.g., K-Means), Spectral Clustering captures the global connectivity structure of the data manifold, making it more suitable for identifying semantic consistency in self-supervised representations.

#### 4.2.2. ANCHOR SELECTION

After clustering $\mathcal{Z}_{aux}$ into $K$ clusters, we implement a *Cluster Filtering* strategy to identify optimal target regions. We prioritize clusters whose size exceeds a predefined den-

sity threshold $\gamma_{ach}$ (i.e., $|C_k| > \gamma_{ach}$). We argue that larger clusters represent universal feature manifolds that capture fundamental semantic patterns learned by the encoder. Sparse clusters often correspond to outliers or fragile, high-frequency features that are likely to be discarded or drastically altered during the attacker's downstream fine-tuning. By guiding fingerprints into these dense, stable regions, we ensure higher robustness against model modification. Upon filtering, the defender selects a target cluster $C_{tgt}$. From $C_{tgt}$, we randomly sample $M$ embeddings to serve as *Anchors*, denoted as $\mathcal{A} = \{\mathbf{a}_1, \mathbf{a}_2, \ldots, \mathbf{a}_M\}$. These anchors represent the destination in the feature space for our fingerprint migration.

#### 4.2.3. ADVERSARIAL SHIFTING

We define the *Group Fingerprint* by selecting $N$ base samples $\{x'_1, x'_2, \ldots, x'_N\}$ from other clusters ($\mathcal{C}_k \neq \mathcal{C}_{tgt}$). Our goal is to perturb these samples such that their outputs from $E_{vct}$ shift towards the target anchors $\mathcal{A}$. We formulate this as an optimization problem called **Adversarial Shifting**:

$$\min_{\delta_j} \mathcal{L}_{shift} = \sum_{i=1}^{N} \|E_{vct}(x'_j + \delta_j) - \mathbf{a}_i\|_2^2, \qquad (5)$$
$$\text{s.t. } \|\delta_j\|_2 \leq \epsilon$$

where $\delta_j$ is the adversarial perturbation added to the $j$-th base sample, and the perturbation budget $\epsilon$ is the $\ell_2$-norm constraint used to maintain the perceptual quality of the samples. We solve this using Projected Gradient Descent (PGD):

$$\delta_j = -\alpha \cdot \nabla_{x_j} \mathcal{L}_{shift}$$
$$x_j^{(t+1)} = \text{Proj}_\epsilon \left( x_j^{(t)} + \delta_j \right), \qquad (6)$$

where $\alpha$ is the step size and $\text{Proj}_\epsilon(\cdot)$ projects the perturbation back into the $\epsilon$-ball. This results in the group fingerprint

$\mathcal{G} = \{x_{mem}^{(i)}\}_{i=1}^{N}$, which is composed of $N$ members. The detailed procedure for fingerprint generation is summarized in Algorithm 1 (Appendix). By minimizing the distance to the anchors, we effectively force these members to aggregate into a "feature island" that is distinct from their original semantic regions. Because these members are extremely close to each other in the feature space of the victim encoder $E_{vct}$, any downstream classifier $H_{spc}$—which essentially partitions the feature space—will almost certainly assign the entire group fingerprint to the same (though unknown) label. This intra-group consistency serves as the immutable marker of the encoder's identity.

### 4.3. Ownership Verification

Once the group fingerprints are generated, the defender performs ownership verification by querying the black-box API of the *Suspect Model* $f_{spc}$. The verification process is designed to be label-only and downstream-agnostic, requiring no access to the suspect model's internal embeddings or specific task semantics. To overcome this, we propose a *Group Voting* mechanism that exploits the intra-group consistency of our generated fingerprints.

Since members of a group fingerprint $\mathcal{G} = \{x_{mem}^{(i)}\}_{i=1}^{N}$ are tightly clustered in the feature space of $E_{vct}$, they are expected to fall within the same decision region of any subsequent classifier head. Consequently, they are expected to yield consistent predictions from $f_{spc}$. We quantify this consistency by identifying the majority label $\hat{y}$ and calculating the *Voting Ratio* $r_{vot}$:

$$\hat{y} = \arg\max_{c} \sum_{x \in \mathcal{G}} \mathbb{I}(f_{spc}(x) = c),$$

$$r_{vot}(\mathcal{G}) = \frac{1}{N} \sum_{i=1}^{N} \mathbb{I}(f_{spc}(x_{mem}^{(i)}) = \hat{y}), \tag{7}$$

where $c$ denotes a candidate class label within the unknown downstream task's output space. Algorithm 2 in the Appendix delineates the proposed group voting mechanism. A group $\mathcal{G}$ is deemed a valid fingerprint match if its consistency exceeds a *Voting Threshold* $\gamma_{vot}$ (default $\gamma_{vot} = 0.5$). We define the group verification indicator as $\Phi(\mathcal{G}) = \mathbb{I}(r_{vot}(\mathcal{G}) > \gamma_{vot})$.

To mitigate the risk of accidental collisions in verification, the defender employs a set of $L$ disjoint group fingerprints $\{\mathcal{G}_l\}_{l=1}^{L}$. Final ownership is determined by a metric reflecting the alignment of fingerprints with the suspect model, such as the *Matching Rate* (MR):

$$\text{MR} = \frac{1}{L} \sum_{l=1}^{L} \Phi(\mathcal{G}_l). \tag{8}$$

If MR exceeds a pre-determined *Verification Threshold* $\gamma_{vrf}$, we conclude that $f_{spc}$ is an infringing model derived from

$E_{vct}$. This hierarchical aggregation—from member consistency to group success—enables precise IP verification without accessing model embeddings.

## 5. Experiment

In our experiments, we conduct comprehensive evaluations on seven datasets spanning Computer Vision (i.e., CIFAR-10/100 (Krizhevsky et al., 2010), STL-10 (Coates et al., 2011), GTSRB (Stallkamp et al., 2011), ImageNet (Deng et al., 2009)) and Natural Language Processing (i.e., SNLI (Bowman et al., 2015), MRPC (Dolan & Brockett, 2005)). We utilize representative pre-trained encoders, including ResNet-50 (trained via Supervised, SimCLR, and MoCoV2), ViT-B (SigLIP), and BERT, to demonstrate the generalizability of our approach. We compare our method against four state-of-the-art protection schemes: **SSL-WM** (Lv et al., 2024a), **MEA** (Lv et al., 2024b), **ADV-TRA** (Xu et al., 2024), and **MFUE** (Xu et al., 2026). We start by evaluating the verification performance under two common downstream training strategies: FTLL and FTAL. In FTLL, the parameters of the pre-trained encoder are frozen, and only the last classification head is updated. Conversely, FTAL involves updating all model parameters, which presents a more challenging scenario that has been less explored in the literature. To further assess the robustness of our fingerprints under the strict black-box threat model, we subject the victim encoder to diverse attacks, including model fine-tuning, pruning (Han et al., 2015), input/embedding perturbations, and three model extraction attacks (i.e., StolenEncoder (Liu et al., 2022), DFME (Truong et al., 2021), and DFMS-HL (Sanyal et al., 2022)). Complete experimental details, including specific implementation parameters, network architectures, and metrics, are provided in Appendix D.

Furthermore, we evaluate the stealthiness of our generated fingerprint samples by testing their resistance against query-side detection mechanisms, as detailed in Appendix E.1. We also provide a thorough analysis of our design components through extensive ablation studies in Appendix E.2.

### 5.1. Main Performance

In this section, we evaluate the uniqueness, fidelity, universality, and feasibility of our proposed fingerprinting scheme across diverse pre-training paradigms (SimCLR, MoCoV2, SigLIP, and Supervised Learning) and downstream tasks (CV and NLP). The main results are reported in Table 1.

We first assess the verification performance under different downstream adaptation strategies. In the standard setting where only the classification head is re-trained (FTLL), our method achieves near-perfect verification performance (Matching Rate > 0.9 in most cases) across all datasets.

*Table 1.* Main performance comparison with four state-of-the-art methods across various pre-training paradigms and downstream tasks. We report the Clean Data Accuracy (CDA) and Matching Rate under two downstream adaptation strategies: FTLL and FTAL.

| Training Method | Downstream Task | CDA (FTLL / FTAL) | | | Matching Rate (FTLL / FTAL) | | | | |
|---|---|---|---|---|---|---|---|---|---|
| | | Clean | SSL-WM | MEA | SSL-WM | MEA | ADV-TRA | MFUE | Ours |
| **SimCLR** | STL-10 | 0.791/0.834 | 0.776/0.819 | 0.783/0.830 | 0.357/0.298 | –/0.751 | 0.323/0.287 | 0.426/0.392 | **0.960/0.760** |
| | GTSRB | 0.772/0.895 | 0.763/0.884 | 0.760/0.882 | 0.583/0.466 | –/0.623 | 0.396/0.336 | 0.403/0.354 | **0.960/0.800** |
| | CIFAR-100 | 0.460/0.473 | 0.439/0.454 | 0.457/0.466 | 0.484/0.291 | –/0.601 | 0.280/0.230 | 0.320/0.279 | **0.920/0.660** |
| **MoCoV2** | STL-10 | 0.880/0.839 | 0.862/0.835 | 0.871/0.837 | 0.462/0.763 | –/0.717 | 0.341/0.292 | 0.451/0.406 | **1.000/0.800** |
| | GTSRB | 0.862/0.927 | 0.853/0.921 | 0.858/0.928 | 0.465/0.349 | **–/0.798** | 0.458/0.379 | 0.428/0.378 | **0.940/0.740** |
| | CIFAR-100 | 0.477/0.483 | 0.457/0.465 | 0.466/0.478 | 0.540/0.651 | –/0.589 | 0.263/0.218 | 0.337/0.295 | **0.900/0.680** |
| **SigLIP** | STL-10 | 0.776/0.801 | – | – | – | – | 0.318/0.277 | 0.389/0.353 | **0.920/0.760** |
| | GTSRB | 0.754/0.835 | – | – | – | – | 0.385/0.331 | 0.365/0.341 | **0.940/0.720** |
| | CIFAR-100 | 0.373/0.419 | – | – | – | – | 0.254/0.208 | 0.256/0.226 | **0.940/0.640** |
| **Supervised Learning** | STL-10 | 0.821/0.892 | 0.803/0.883 | 0.813/0.890 | 0.628/0.470 | –/0.730 | 0.307/0.269 | 0.434/0.390 | **1.000/0.840** |
| | GTSRB | 0.886/0.983 | 0.873/0.970 | 0.882/0.971 | 0.231/0.168 | –/0.745 | 0.475/0.400 | 0.413/0.359 | **1.000/0.820** |
| | CIFAR-100 | 0.492/0.533 | 0.472/0.516 | 0.482/0.529 | 0.355/0.230 | –/0.629 | 0.279/0.229 | 0.330/0.288 | **0.960/0.780** |
| | SNLI | 0.788/0.838 | 0.770/0.812 | 0.766/0.830 | 0.132/0.101 | –/0.662 | – | – | **0.860/0.700** |
| | MRPC | 0.693/0.740 | 0.687/0.736 | 0.690/0.729 | 0.207/0.155 | –/0.576 | – | – | **0.820/0.640** |
| **Additional Knowledge** | – | – | – | – | Output Logits | Encoder Output | Downstream Dataset&Head | Downstream Dataset&Head | **None** |

1 MEA scores 1.0 under FTLL, a trivial result since it depends directly on the frozen encoder's output, rendering this comparison meaningless.
2 "–" denotes that the method is not applicable to the specific model architecture or task.

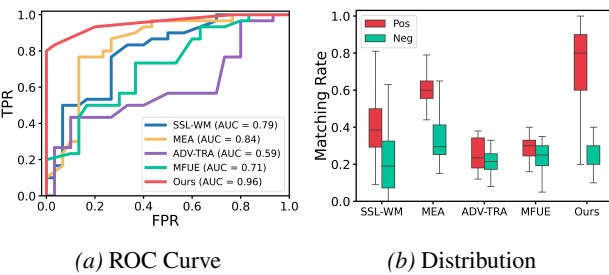

| | (a) ROC Curve | | (b) Distribution |

*Figure 4.* On CIFAR-100, we present (a) the ROC curve, (b) the distribution of positive and negative models.

*Table 2.* Average time required from watermarking/fingerprinting sample generation to verification.

| | STL-10 | GTSRB | CIFAR-100 |
|---|---|---|---|
| FTLL | 16m | 40m | 61m |
| FTAL | 41m | 100m | 152m |
| SSL-WM | 204m | 427m | 681m |
| MEA | 124m | 141m | 203m |
| ADV-TRA | **7m** | **8m** | **10m** |
| MFUE | 51m | 89m | 124m |
| **Ours** | 10m | 12m | 17m |

In contrast, baseline methods like ADV-TRA and MFUE exhibit significantly lower matching rates (typically $< 0.5$). This is because these methods rely on downstream task-specific decision boundaries, which are completely reconstructed by the new classification head in the "Pre-trained Encoder + Downstream Head" paradigm. Furthermore, we evaluate the robustness under a more challenging scenario: full model fine-tuning (FTAL). This process significantly alters the encoder's parameters and feature space, causing catastrophic performance drops in baselines (e.g., SSL-WM drops to 0.291 on CIFAR-100). Conversely, our method experiences only a minor degradation; for instance, on the SimCLR encoder transferred to GTSRB, we sustain a 0.8 matching rate. This resilience validates that by shifting samples to robust clusters in the feature space, our fingerprint becomes an intrinsic property of the encoder that persists even when parameters are perturbed.

While achieving high effectiveness is crucial, it must not come at the expense of model utility. As shown in the comparison, watermarking methods such as SSL-WM and MEA inevitably degrade the CDA. Even numerically marginal losses—averaging 0.41% for SSL-WM—are increasingly intolerable in the pursuit of state-of-the-art precision. In

contrast, fingerprinting schemes (ADV-TRA, MFUE, and our method) maintain the original performance without any interference. This highlights the superior ***fidelity*** of our approach, as it enables robust ownership verification through a non-invasive paradigm that leaves the encoder's intrinsic utility untouched.

A reliable fingerprint must accurately distinguish between stolen and innocent models. Figure 4a compares the ROC curves and AUC scores. Our method achieves a superior AUC of 0.96, significantly outperforming the second-best MEA (0.84) and far surpassing ADV-TRA (0.59) and MFUE (0.71). To further analyze the discriminative margin, we visualize the matching rate distributions in Figure 4b. While baselines suffer from significant overlaps or high variance, our method demonstrates a distinct separation (positive $\sim 0.8$ vs. negative $\sim 0.25$). This substantial margin validates the superior ***uniqueness*** of our feature-space fingerprint, confirming it is robustly preserved during downstream adaptation while remaining strictly distinguishable from innocent models.

To further assess the generalization capability of our method, Figure 5 illustrates the matching rates on positive models derived via FTAL across varying model capacities, ranging from lightweight CNNs to heavy ViTs. Regardless of the significant differences in parameter scales and architec-

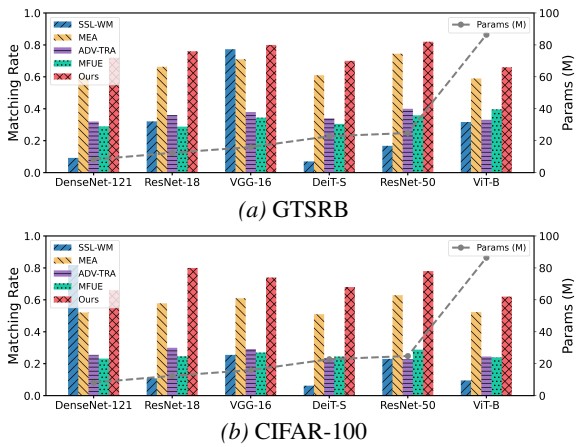

*Figure 5.* Matching rates and parameter counts of positive models (derived via FTAL) evaluated on four distinct CNN architectures and two Vision Transformer variants.

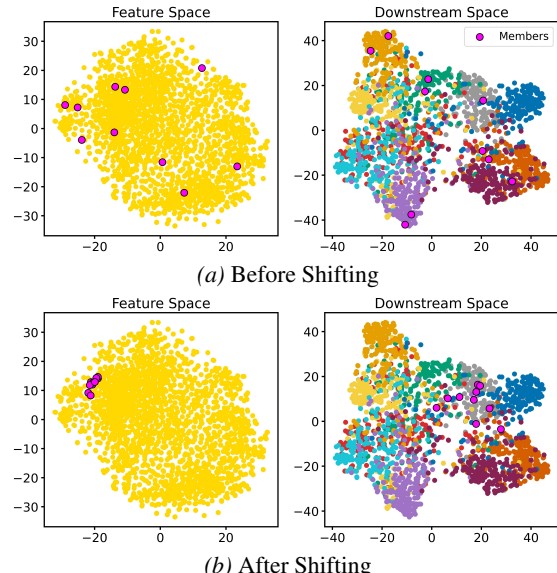

*Figure 6.* t-SNE visualization of the feature and downstream spaces on STL-10, comparing distributions before (a) and after (b) adversarial shifting. Colors in the downstream space represent the model-predicted classes.

tural inductive biases, our method consistently maintains high verification accuracy across both GTSRB and CIFAR-100 datasets. For instance, the matching rates on ViTs are comparable to those on ResNets, indicating that our feature-space fingerprint is not coupled to specific local structures (e.g., convolution kernels). Combined with the results in Table 1, which verify effectiveness across diverse pre-training paradigms (e.g., self-supervised vs. supervised), these findings collectively demonstrate the extensive *universality* of our fingerprinting framework across heterogeneous model architectures and training mechanisms.

A critical advantage of our framework is its strict adherence to the label-only and downstream-agnostic threat model. As summarized in the bottom row of Table 1, existing baselines rely on unrealistic assumptions for verification in this scenario. SSL-WM requires access to output logits to compute entropy; MEA requires direct access to the encoder's output embeddings; ADV-TRA and MFUE require prior knowledge of the downstream dataset to train surrogate models. Conversely, our method requires zero additional knowledge. Furthermore, we evaluate the time consumption in Table 2. Watermarking approaches (e.g., SSL-WM and MEA) incur prohibitive time costs (up to 681 minutes) due to the necessity of training surrogate/shadow encoders. In contrast, our method is surrogate-free, completing the process within $\sim 17$ minutes on CIFAR-100. By leveraging the "Group Voting" mechanism to enable label-only verification with minor computational overhead, our approach satisfies strict deployment constraints, demonstrating superior *feasibility* in real-world EaaS scenarios.

## 5.2. Visualization of Feature Space Shifting

To intuitively verify our core hypothesis—that feature-space clustering implies downstream consistency—we visualize the embedding distributions on STL-10 via t-SNE in Figure 6. Initially (Figure 6a), the member samples are dispers-

edly distributed across the feature manifold, resulting in divergent predictions in the downstream space. However, after applying adversarial shifting (Figure 6b), these samples are successfully aggregated into a compact and isolated cluster within the encoder's feature space. Crucially, this intrinsic cohesion effectively propagates to the downstream space, where the samples remain tightly grouped and are mapped to a specific class region (colored with gray). This phenomenon corroborates our design philosophy: *by constructing a distinct "feature island" within the encoder's output space, we decouple the fingerprint from specific decision boundaries*. This ensures that the fingerprint remains robust and identifiable (via Group Voting) regardless of how the downstream head reconstructs the semantic space, thereby realizing our downstream-agnostic goal.

## 5.3. Robustness Evaluation

In this section, we systematically evaluate the robustness of fingerprinting and watermarking methods against two categories of threats: **model modification** (including fine-tuning, pruning, and model extraction) and **inference-time perturbation** (including input processing and embedding perturbing). To ensure a fair and standardized comparison, all experiments are conducted on the SimCLR-GTSRB and MoCoV2-CIFAR-100 benchmarks under the FTLL setting (consistent with Table 1).

**Fine-tuning**. As a common way of model modification, fine-tuning aggressively alters encoder weights and reshapes the feature space. Figure 7a and 7b illustrate the matching rate

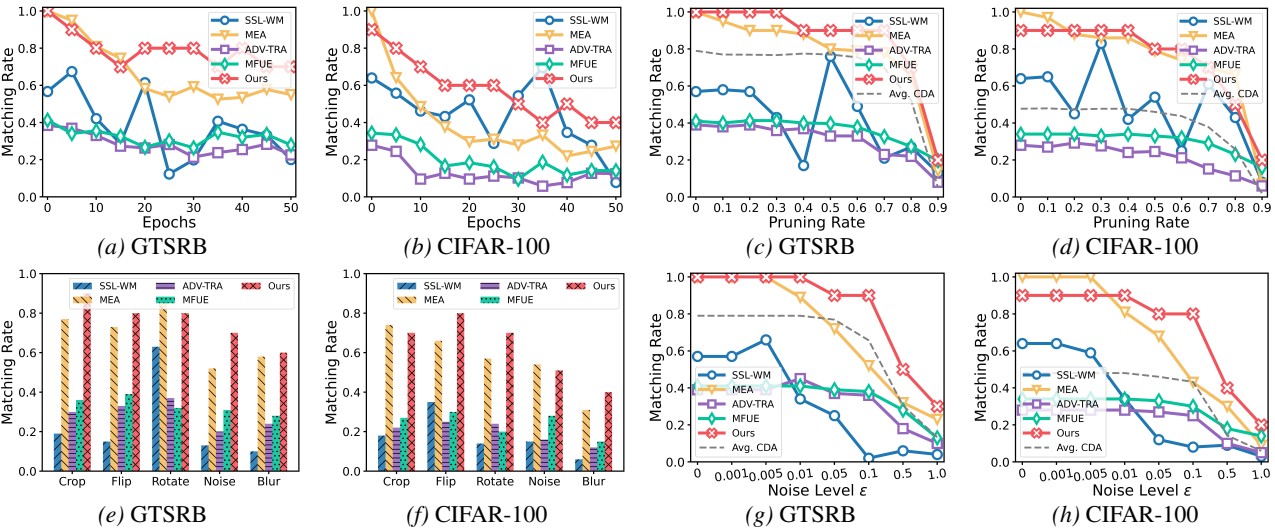

*Figure 7.* Matching rates under four types of attacks, including fine-tuning (a,b), pruning (c,d), input processing (e,f), and embedding perturbing (g,h).

dynamics over 50 epochs. As illustrated, baseline methods struggle significantly under this attack. Fingerprinting methods (e.g., ADV-TRA, MFUE) hover at low matching rates ($< 0.4$) throughout the process, while SSL-WM exhibits rapid decay (on CIFAR-100) with extreme instability. Notably, MEA performs relatively well among the baselines. We attribute this to its verification mechanism, which operates directly on encoder embeddings and bypasses the downstream head—a component that is far more volatile than the encoder during fine-tuning. In contrast, our method demonstrates remarkable resilience. On GTSRB, our matching rate remains consistently above $0.7$, and even on the more challenging CIFAR-100 dataset, it outperforms the nearest competitor by a clear margin ($\sim 10\% - 20\%$) in the later stages of training.

**Pruning.** We further evaluate robustness against model compression by varying the pruning sparsity ratio from 0 to 0.9, as shown in Figure 7c and 7d. The grey dashed line represents the average CDA, which naturally declines as the model is compressed. As observed, baseline methods struggle to maintain effectiveness; ADV-TRA and MFUE consistently yield low matching rates ($< 0.4$), while SSL-WM exhibits extreme volatility. Conversely, our method demonstrates superior robustness, maintaining a high matching rate ($> 0.9$) even when 60%-70% of the parameters are pruned. A significant performance drop is observed only when the pruning rate exceeds $0.8$, coinciding with the collapse of the model's utility (where CDA drops near zero). This indicates that our fingerprint persists as long as the model remains functional. We attribute this robustness to our strategy of anchoring fingerprints within the feature space structure; unlike parameter-specific or boundary-sensitive methods, the clusters formed by our adversarial shifting are preserved

even after removing redundant weights.

**Input Processing.** Turning to inference-time perturbations, we first evaluate robustness against five common image transformations (Cropping, Flipping, Rotating, Gaussian Noise, and Gaussian Blur), as shown in Figure 7e and 7f. Observations indicate that standard fingerprinting methods (ADV-TRA, MFUE) and the logit-based watermark (SSL-WM) exhibit significant fragility. For instance, on GTSRB, these methods generally fail to maintain a matching rate above $0.4$, while MEA exhibits a noticeable performance decline under noise and blur. In contrast, our method maintains superior robustness across all scenarios. Specifically, for geometric transformations on GTSRB, our matching rate remains consistently above $0.8$. Even under severe corruptions like Noise and Blur, our method outperforms competitors by a significant margin (e.g., exceeding MEA by $0.18$ under Noise). This is likely because our group voting mechanism relies on the consensus of a sample group rather than fragile single triggers used by baselines. Even if input processing causes random prediction errors for a few individual samples, the aggregated result of the group remains stable.

**Embedding Perturbing.** We also asses robustness against Gaussian noise added to encoder outputs, with results shown in Figure 7g and 7h. As the noise level $\epsilon$ increases, the model's utility (grey dashed line) naturally declines. Notably, MEA exhibits unexpected fragility in this scenario compared to previous attacks, with its matching rate dropping significantly (e.g., to $0.43$ on CIFAR-100 at $\epsilon = 0.1$). This vulnerability likely arises because MEA relies on precise point-to-point mapping of watermark samples to target representations; direct feature perturbations easily disrupt

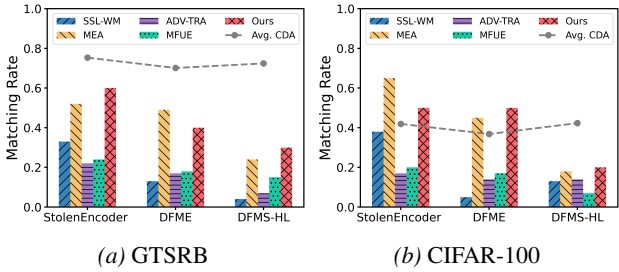

*(a) GTSRB*      *(b) CIFAR-100*

*Figure 8.* Matching rates of surrogate models derived via three model extraction attacks.

this specific mapping, causing verification failure. In contrast, our method demonstrates superior tolerance, maintaining high matching rates ($> 0.8$) even under significant noise levels where baselines falter. This robustness is largely attributed to our group voting mechanism. By aggregating predictions from a cluster of samples rather than relying on individual embeddings, the stochastic effects of the noise are statistically mitigated. Consequently, our fingerprint remains detectable until the noise becomes destructive enough to compromise the model's fundamental utility.

**Model Extraction.** Figure 8 reports the matching rates under StolenEncoder, DFME, and DFMS-HL attacks. Compared to the previous modifications attacks, model extraction generally poses a more severe threat, inducing a sharper decline in matching rates. Among these, StolenEncoder allows for the highest fingerprint retention, as it directly distills the encoder's output embeddings, thereby maximally preserving the feature space structure where our method reside (e.g., reaching 0.6 on GTSRB). While MEA also shows resilience here due to its encoder-level verification, the other three baselines struggle significantly. Notably, DFMS-HL attack presents the most challenging scenario, causing a substantial performance drop for all methods (matching rate $< 0.3$). We attribute this universal failure to the hard-label extraction setting: unlike StolenEncoder (embeddings) or DFME (soft labels), DFMS-HL trains surrogates solely on discrete class predictions. This quantization strips away the subtle probability distributions and gradient information required to transfer fingerprints/watermarks, preventing them from being learned by the surrogate model. Despite this severe information loss, our method maintains a comparative advantage, demonstrating the superior persistence of our cluster-based design compared to boundary-dependent methods.

## 6. Conclusion

In this paper, we presented a novel fingerprinting framework designed to protect the intellectual property of pre-trained encoders, specifically addressing the challenges posed by unknown downstream tasks and black-box, label-only ac-

cess. Unlike conventional methods that rely on fixed end-to-end mappings or require access to internal embeddings, we exploited the layer-wise accumulation of adversarial perturbations. By introducing an unsupervised anchor-guided adversarial shifting mechanism, we successfully migrated fingerprint samples into cohesive clusters within the feature space, establishing robust, endogenous fingerprints. Furthermore, our proposed group voting mechanism effectively circumvents the issue of semantic reconstruction caused by downstream fine-tuning, enabling precise ownership verification without prior knowledge of the downstream task. Extensive experiments demonstrate that our approach is highly robust against various model extraction attacks, including fine-tuning and pruning. Beyond serving as a universal defense tool for high-value foundation models, this work offers a new perspective on controlling the transferability of adversarial examples within feature manifolds, paving the way for future research in trustworthy deep learning.

## Acknowledgements

This work was supported in part by the National Natural Science Foundation of China under Grants 62506138, 62272183, 62071192 and 62502477; by the National Key R&D Program of China under Grant 2024YFE0103800; by the Key R&D Program of Hubei Province under Grants 2025EHA033, 2024BAB016 and 2024BAB031; by the Major Science and Technology Project of Hubei Province under Grants 2024BAA008 and 2024BAA011; and by the Open Project Funding of the Key Laboratory of Intelligent Sensing System and Security, Ministry of Education under Grant KLISSS202401.

## Impact Statement

This work advances machine learning by establishing a robust ownership verification mechanism for pre-trained encoders. Since these encoders are high-value intellectual property requiring immense training costs in data and computation, our research provides essential protection in black-box and downstream-agnostic settings. This fosters an accountable AI ecosystem and supports the sustainable development of foundation models by safeguarding substantial research and development investments. The proposed fingerprinting is non-intrusive and maintains model utility without introducing biases. We anticipate no negative societal consequences, as this study primarily offers defensive safeguards for the AI community.

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

# Appendix

In this appendix, we provide supplementary materials to further support the findings and technical details discussed in the main text. The contents are organized as follows:

- **Appendix A: Extended Related Work**. Presents a comprehensive literature review focusing on the evolution of IP protection for deep neural networks and the specific challenges in protecting pre-trained encoders.
- **Appendix B: Discussion**. Discusses the limitations of the proposed framework, including NLP modality challenges, extreme model modifications, and scalability to large output spaces, alongside potential avenues for future research.
- **Appendix C: Formal Algorithms and Pseudocode**. Detailed procedural descriptions for *Adversarial Shifting* (Generation Phase) and *Group Voting* (Verification Phase).
- **Appendix D: Detailed Experimental Setup**. Comprehensive information regarding datasets, model architectures (e.g., ResNet, ViT), hyperparameter configurations, and training schedules.
- **Appendix E: Additional Experimental Results**. Supplementary evaluations including the stealthiness (perceptual quality) of fingerprints, uniqueness analysis against independent encoders, and extensive ablation studies on key parameters.
- **Appendix F: Observation**. Visualizes and analyzes the cumulative deviation of adversarial samples across different layers of the encoder, providing empirical justification for our feature-space shifting strategy.

## A. Related Work

### A.1. Model Fingerprinting

Model fingerprinting serves as a post-hoc verification mechanism to determine whether a suspect model is an illegal copy of a victim model (Lederer et al., 2024; Ye et al., 2025). Early pioneering works, such as IPGuard (Cao et al., 2021) and Conferrable Examples (Lukas et al., 2021), leverage the transferability of adversarial examples to construct fingerprints. These methods generate specific inputs near the decision boundary that elicit incorrect predictions from the victim model. If the suspect model exhibits identical misclassification behaviors on these inputs, it is identified as a surrogate.

To counter adaptive defenses and model modifications (e.g., fine-tuning, pruning), subsequent research focused on enhancing fingerprint robustness. Yang et al. (2022) utilize meta-learning to fingerprint the model's inner decision areas rather than fragile boundaries. Other approaches explored global characteristics: UAP (Peng et al., 2022) exploits universal adversarial perturbations to capture subspace profiles, while SAC (Guan et al., 2022) and specialized verified frameworks [8] utilize sample correlations and unique verification protocols to detect theft. Similarly, ADV-TRA (Xu et al., 2024) introduces adversarial trajectories to characterize the progressive changes across decision boundaries, offering greater stability than single-point samples. More recently, MFUE (Xu et al., 2026) utilizes unlearnable examples to fingerprint the model in the parameter space, circumventing boundary sensitivity altogether. To systematize these diverse methodologies, Godinot et al. (2025) propose the QuRD framework, offering a unified evaluation standard and revealing that simple baselines can sometimes rival complex state-of-the-art schemes.

Despite their effectiveness in end-to-end scenarios, these methods predominantly rely on the assumption that the suspect model shares the same category semantics and output space as the victim. They fundamentally depend on verifying the "input-label" mapping consistency. However, in the context of pre-trained encoders, the downstream head is often replaced, altering the output dimension and semantic meaning. This disruption of the output space renders traditional label-matching fingerprints ineffective, highlighting the need for verification mechanisms that are agnostic to downstream tasks.

### A.2. Pre-trained Encoder IP Protection

As pre-trained encoders (e.g., self-supervised models) become foundational to modern AI, their IP value has surged. Liu et al. first demonstrate the severity of this threat by proposing StolenEncoder (2022), an attack that effectively steals the functionality of self-supervised encoders, highlighting the urgent need for robust protection mechanisms.

In response, recent research has pivoted from protecting end-to-end models to safeguarding pre-trained encoders. The dominant paradigm is active watermarking, which embeds verifiable patterns into the encoder during the training phase. Cong et al. propose SSLGuard (2022), a pioneering framework that injects watermarks into encoders by defining multiple objective losses. Subsequent works focus on enhancing the robustness and applicability of these watermarks. SSL-WM (Lv et al., 2024a) extends protection to black-box settings, designing watermarks intended to persist through downstream

*Table 3.* Comparison of our proposed method with representative model IP protection schemes. Our method is the first to achieve encoder-level ownership verification under the challenging downstream-agnostic, and label-only setting.

| Method | Target | Accuracy-preserving | Surrogate-free | Downstream-agnostic | Require |
|---|---|---|---|---|---|
| IPGuard ADV-TRA | Full Model | ✓ | ✓ | ✗ | **Labels** |
| MFUE | Full Model | ✓ | ✗ | ✗ | **Labels** |
| SSLGuard | Encoder | ✗ | ✗ | ✗ | Embeddings |
| SSL-WM | **Encoder & Full Model** | ✗ | ✓ | ✗ | Logits |
| MEA | **Encoder & Full Model** | ✗ | ✓ | ✗ | Embeddings |
| StegGuard | Encoder | ✓ | ✗ | ✗ | Embeddings |
| **Ours** | **Encoder & Full Model** | ✓ | ✓ | ✓ | **Labels** |

fine-tuning and manifest in the final predictions. MEA (Lv et al., 2024b) further improve robustness against model extraction attacks by using synthetic trigger samples. The scope of encoder protection has also diversified: BlackMark (Li et al., 2025) explores commercial-grade black-box watermarking schemes. Addressing other modalities of pre-trained encoders, PreGIP (Dai et al., 2025) extends these concepts to graph neural networks (GNNs). As the first fingerprinting scheme tailored for encoders, StegGuard (Ren et al., 2025) leverages steganographic principles to treat the encoder and decoder as a secret key pair.

However, these existing solutions face several fundamental limitations. First, they are predominantly invasive watermarking techniques. They require the defender to intervene in the computationally expensive pre-training process (e.g., modifying loss functions or data), which inevitably introduces inductive bias and may degrade the encoder's utility on clean data. Moreover, to ensure robustness, some methods rely on training surrogate (also referred to as shadow or mimic) models to simulate the behavior of tampered models. These surrogates are then utilized to optimize the trigger samples required for verification, which imposes significant computational overhead. Besides, they often struggle with the verification dilemma in realistic APIs. Many methods (e.g., SSLGuard) rely on accessing the encoder's output embeddings for verification. Even SSL-WM (Lv et al., 2024a), which addresses the most practical scenarios to date, still necessitates access to the logits (soft labels) output by the downstream head. In stricter scenarios where only the final downstream labels are exposed, and the downstream task semantics are unknown (downstream-agnostic), verifying a watermark defined in the latent space becomes intractable. Unlike these methods, our approach proposes the first post-hoc fingerprinting mechanism for encoders, requiring no training intervention and enabling robust verification strictly via label-only access. Specifically, we provide a comprehensive comparison with representative state-of-the-art methods across four dimensions—expansibility, fidelity, and verification constraints—as summarized in Table 3.

## B. Discussion

While our proposed framework utilizing Adversarial Shifting and Group Voting provides a highly robust, downstream-agnostic ownership verification mechanism for pre-trained encoders, we identify several limitations and important avenues for future research:

**NLP Modality Challenges and Input Reconstruction.** Extending our adversarial shifting mechanism to discrete modalities, such as NLP, introduces implementation complexities absent in continuous vision domains. Following established protocols, our current NLP evaluation is primarily conducted within the continuous embedding space of the encoder (e.g., BERT), utilizing PGD bounded by an $l_2$-norm constraint. While practical black-box deployment necessitates projecting these continuous embedding perturbations back into valid, stealthy discrete token sequences, this introduces an additional reconstruction overhead. Although this can be effectively mitigated using constrained Nearest Neighbor Search (NNS) paired with part-of-speech and semantic similarity checks—a well-established technique in recent adversarial NLP literature—the absence of an end-to-end discrete token mapping in our current pipeline remains a limitation. We emphasize that this embedding-to-token reconstruction is orthogonal to our core contribution of feature-space aggregation, yet it constitutes a vital step for robust real-world NLP verification.

**Extreme Model Modifications.** Our empirical evaluations demonstrate that the proposed fingerprinting method is highly

resilient to standard fine-tuning, pruning, and embedding perturbations. However, highly destructive attacks—such as hard-label Data-Free Model Stealing (e.g., DFMS-HL) or extreme parameter pruning (e.g., $> 80\%$)—can severely distort the intrinsic structure of the encoder's feature space, inevitably leading to a degradation in the matching rate. It is important to note, however, that such extreme modifications concurrently destroy the model's inherent utility, rendering the stolen model practically unusable for the attacker. Thus, a natural robustness-utility trade-off exists, and our fingerprint remains reliably verifiable as long as the stolen model retains meaningful functional value.

**Scalability to Large Output Spaces.** Our label-only verification relies on the consensus of fingerprint members within the downstream prediction space. For downstream tasks characterized by an exceptionally large number of target classes (e.g., $> 100$), the verification margin narrows slightly. In such highly granular output spaces, it becomes statistically more challenging for the entire "feature island" to seamlessly map to a single class boundary. While the overall effectiveness of our framework remains intact, this necessitates careful calibration of the voting threshold ($\gamma_{vot}$) to maintain optimal verification confidence.

# C. Pseudocode

---

**Algorithm 1** Group Fingerprint Generation via Adversarial Shifting

---

**Input:** Victim Encoder $E_{vct}$, Auxiliary Dataset $\mathcal{D}_{aux}$, Density Threshold $\gamma_{ach}$, Number of Anchors $M$, Group Size $N$, Perturbation Budget $\epsilon$, Step Size $\alpha$, Iterations $T$.
**Output:** Group Fingerprint $\mathcal{G}$.
**// Phase 1: Feature Space Characterization**
Extract features: $\mathcal{Z}_{aux} \leftarrow \{E_{vct}(x) \mid x \in \mathcal{D}_{aux}\}$.
Partition $\mathcal{Z}_{aux}$ into clusters $\mathcal{C} = \{C_1, \ldots, C_K\}$ via Spectral Clustering.
Filter large clusters: $\mathcal{C}_{large} \leftarrow \{C_k \in \mathcal{C} \mid |C_k| > \gamma_{ach}\}$.
Initialize group fingerprint $\mathcal{G} \leftarrow \emptyset$.
**// Phase 2: Anchor Selection**
Select a target cluster $C_{tgt}$ from $\mathcal{C}_{large}$.
Randomly sample $M$ anchors $\mathcal{A} = \{\mathbf{a}_1, \ldots, \mathbf{a}_M\}$ from $C_{tgt}$.
**// Phase 3: Adversarial Shifting**
Select $N$ base samples $X = \{x'_1, \ldots, x'_N\}$ from $\mathcal{C} \setminus C_{tgt}$.
Initialize perturbations $\delta_j \leftarrow 0$ for $j = 1 \ldots N$.
**for** $j = 1$ **to** $N$ **do**
    **for** $t = 1$ **to** $T$ **do**
        Calculate loss $\mathcal{L}_{shift}$:
            $\mathcal{L}_{shift} = \sum_{i=1}^{M} \|E_{vct}(x_j^{(t)} + \delta_j) - \mathbf{a}_i\|_2^2$
        Update members using PGD:
            $\delta_j \leftarrow -\alpha \cdot \nabla_{x_j} \mathcal{L}_{shift}$
            $x_j^{(t+1)} \leftarrow \text{Proj}_\epsilon(x_j^{(t)} + \delta_j)$
    **end for**
    $\mathcal{G} \leftarrow x_j^{(T)}$.
**end for**
**return** $\mathcal{G} = \{x_{mem}^j\}_{j=1}^N$

---

---

**Algorithm 2** Group Voting

---

**Input:** Suspect Model $f_{spc}$, Group Fingerprint $\mathcal{G}$, Voting Threshold $\gamma_{vot}$.
**Output:** Voting Ratio $r_{vot}$.
Let current group $\mathcal{G} = \{x_{mem}^{(1)}, \ldots, x_{mem}^{(N)}\}$.
Query suspect model to get labels: $Y \leftarrow \{f_{spc}(x) \mid x \in \mathcal{G}\}$.
Identify majority label $\hat{y}$:
$\quad \hat{y} \leftarrow \arg\max_{c} \sum_{x \in \mathcal{G}} \mathbb{I}(f_{spc}(x) = c)$
Calculate voting ratio $r_{vot}$:
$\quad r_{vot} \leftarrow \frac{1}{N} \sum_{i=1}^{N} \mathbb{I}(f_{spc}(x_{mem}^{(i)}) = \hat{y})$
**return** $r_{vot}$

---

# D. Experiment Setup

**Datasets.** We conduct extensive evaluations on seven benchmark datasets, encompassing both Computer Vision (CV) and Natural Language Processing (NLP) tasks. Specifically, for CV tasks, we employ five widely recognized image datasets: CIFAR-10 (Krizhevsky et al., 2010), CIFAR-100 (Krizhevsky et al., 2010), STL-10 (Coates et al., 2011), GTSRB (Stallkamp et al., 2011), and ImageNet (Deng et al., 2009). Note that we resize the images from these datasets to $3 \times 224 \times 224$. In addition, we adopt SNLI (Bowman et al., 2015) and MRPC (Dolan & Brockett, 2005) to evaluate the performance on NLP tasks. The details of these datasets are described as follows.

- **CIFAR-10** (Krizhevsky et al., 2010). This dataset consists of 60,000 color images with a resolution of $32 \times 32$, categorized into 10 classes. It is split into 50,000 training images and 10,000 test images, serving as a standard benchmark for image classification.

- **CIFAR-100** (Krizhevsky et al., 2010). Similar to CIFAR-10, this dataset contains 60,000 images of size $32 \times 32$ but is divided into 100 fine-grained classes. It presents a more challenging classification task due to the limited number of images per class.

- **STL-10** (Coates et al., 2011). Designed for unsupervised and self-supervised learning, STL-10 contains 10 classes of $96 \times 96$ images. It comprises 5,000 labeled training images, 8,000 test images, and importantly, 100,000 unlabeled images derived from a similar distribution.

- **GTSRB** (Stallkamp et al., 2011). The German Traffic Sign Recognition Benchmark (GTSRB) contains 43 distinct classes of traffic signs with over 50,000 images. It is widely used for evaluating model robustness in safety-critical scenarios.

- **ImageNet** (Deng et al., 2009). A large-scale hierarchical image database, widely regarded as the gold standard for pre-training deep learning models. We utilize the ILSVRC-2012 subset, which contains approximately 1.28 million training images spanning 1,000 categories.

- **SNLI** (Bowman et al., 2015). The Stanford Natural Language Inference (SNLI) corpus is a collection of 570k human-written English sentence pairs. It is manually labeled for balanced classification with three labels: entailment, contradiction, and neutral.

- **MRPC** (Dolan & Brockett, 2005). The Microsoft Research Paraphrase Corpus (MRPC) consists of $5,801$ sentence pairs automatically extracted from online news sources. Each pair indicates whether it captures a semantic equivalence/paraphrase relationship.

**Pre-trained Encoder and Downstream task.** For CV tasks, we adopt the ResNet-50 architecture as the target backbone encoder. To ensure a comprehensive evaluation, we utilize encoders pre-trained on the ImageNet dataset across three distinct paradigms—two representative SSL algorithms, SimCLR[1] and MoCoV2[2], and standard supervised learning[3], initializing these models using officially released checkpoints for reproducibility and convenience. In addition to ResNet-50, we

---

[1]https://github.com/google-research/simclr
[2]https://github.com/facebookresearch/moco
[3]https://docs.pytorch.org/vision/stable/

incorporate SigLIP (an efficient approach to image-text contrastive learning) and select its visual encoder component ViT-B[4]. For NLP tasks, we employ the pre-trained BERT base model, accessed via the Hugging Face[5].

For each downstream task, we attach a 3-layer MLP as the classification head, with hidden layers containing 2048, 512, and 256 neurons. We employ STL-10, GTSRB, and CIFAR-100 as downstream datasets for CV tasks, while SNLI and MRPC for NLP tasks. We implement two common fine-tuning strategies for the downstream tasks: FTLL and FTAL. For both strategies, the models are trained for 30 epochs using an Adam optimizer. We assign specific learning rates to each strategy: $5 \times 10^{-3}$ for FTLL and a lower rate of $5 \times 10^{-4}$ for FTAL to ensure training stability. In this work, models incorporating the target encoder are designated as *positive models* (i.e., infringing models) , while those developed independently without the target encoder are referred to as *negative models* (i.e., innocent models).

**Robustness Evaluation.** Adversaries may employ various strategies to tamper with or erase the pre-defined fingerprints/watermarks from the protected encoders. Following previous works, we utilize the model fine-tuned under the FTLL setting as the target model. We evaluate the robustness of our method against two categories of attacks: three *Model Modification Attacks* (i.e., fine-tuning, pruning, and model extraction) and two *Inference-Time Perturbing Attacks* (i.e., input processing and embedding perturbing).

- **Fine-tuning** aims to overwrite the original watermark by continuing to train the model on auxiliary data. We fine-tune the entire target model (including pre-trained encoder) using the downstream datasets. The optimization is performed using an SGD optimizer with a learning rate of $5 \times 10^{-4}$.

- **Pruning** is widely used for model compression, which may inadvertently diminish the model's memory of the embedded fingerprints by removing redundant parameters. Following the technique in (Han et al., 2015), we evaluate robustness by varying the pruning rate $p$ from 0.1 to 0.9. Specifically, we prune a fraction $p$ of the parameters that possess the smallest absolute values.

- **Model Extraction** aims to steal the functionality of a victim model by training a surrogate model using query-response pairs. We employ three attacks with distinct extraction strategies: StolenEncoder (Liu et al., 2022), DFME (Truong et al., 2021), and DFMS-HL (Sanyal et al., 2022). StolenEncoder is specifically designed for stealing SSL encoders. In contrast, DFME and DFMS-HL train a surrogate model by querying samples that approximate the data distribution, utilizing soft labels and hard labels, respectively.

- **Input Processing** applies transformations to query samples to disrupt their verification functionality. We evaluate robustness against five common image processing techniques: cropping, flipping, rotating, Gaussian noising, and Gaussian blurring. Specifically, we add Gaussian noise with standard deviation $\sigma = 16/255$ and apply Gaussian blur with kernel size $k = 5$.

- **Embedding Perturbing** perturbs the encoder outputs by adding Gaussian noise to the extracted embeddings (features), which may also degrade downstream utility. Formally, we add Gaussian noise $\eta \sim \mathcal{N}(0, \epsilon)$ to the embedding vector and vary the noise scale $\epsilon$ from 0.001 to 1.

**Baselines.** We compare our method with four state-of-the-art fingerprinting and watermarking approaches: two encoder-level watermarking methods (**SSL-WM** and **MEA**) and two model-level fingerprinting methods (**ADV-TRA** and **MFUE**).

- **SSL-WM** (Lv et al., 2024a) trains the encoder to map watermarked samples into an invariant representation space, enabling black-box verification through outlier detection of the downstream classifier's output entropy. We construct the shadow dataset by sampling 8,000 images in total from STL-10, GTSRB, and CIFAR-10.

- **MEA** (Lv et al., 2024b) embeds a symbiotic backdoor watermark by constraining watermark samples to lie within the main-task distributions in both the input space and the feature space. We randomly select two classes to synthesize watermarks and adopt the autoencoder-based blending strategy with a blending ratio of 0.5.

- **ADV-TRA** (Xu et al., 2024) utilizes adversarial trajectories to quantify the decision boundaries, which not only tolerates a greater degree of alteration but also reduces the false positives during verification. We set the trajectory length to 40 with $m = 5$.

---

[4] https://huggingface.co/timm/ViT-B-16-SigLIP
[5] https://huggingface.co/google-bert/bert-base-uncased

- **MFUE** (Xu et al., 2026) fingerprints models in the parameter space by leveraging unlearnable examples that maintain consistent loss values across model variants. We adopt the MFUE-w strategy (which assumes access to logits) for verification. We set the number of mimic models to $5$ and the perturbation radius to $8/255$.

It is important to note that our threat model considers a strict black-box scenario where the adversary adapts the stolen encoder to unknown downstream tasks, and the deployed suspect model is queried in a label-only manner. Since the aforementioned baselines cannot be directly applied under such rigorous constraints, we relax the restrictions for them to enable feasible verification. Specifically, for SSL-WM, we assume the defender can access the output logits. For MEA, as it cannot address the downstream-agnostic challenge, we evaluate its protection solely on the encoder level, assuming the defender has access to the feature vectors output by the suspect encoder. For ADV-TRA and MFUE, which are designed for the entire model, we permit the defender to train a surrogate classification head using data from the same downstream task. This head is attached to the encoder to form a complete classification model, upon which the fingerprints are generated for verification.

**Implementation Details.** In our approach, we randomly sample 5000 images from CIFAR-10 for fingerprint generation, encompassing both the fingerprint samples themselves and the samples required for anchor determination. During anchor selection, we employ spectral clustering within the feature space to cluster the inputs, retaining only those clusters containing more than 50 samples. The number of anchors is set to 10. For member optimization, we employ an SGD optimizer with a learning rate of $0.001$ for 200 iterations, imposing an $\ell_2$-norm constraint of $16/255$ on the adversarial perturbations. For NLP evaluations, adversarial shifting is optimized directly within the continuous embedding space rather than at the discrete token level. Following established protocols, this assumes direct access to the suspect model's embedding layer during verification. During verification, the voting threshold is set to $0.5$ for STL-10 and GTSRB, $0.3$ for CIFAR-100. We generate a total of 10 fingerprint groups, with each group comprising 10 members, thereby strictly controlling the total query budget to 100. For fair comparison, SSL-WM, MEA, and MFUE generate the same number of watermark/fingerprint query samples. Since ADV-TRA inherently requires more queries for trajectory-based verification, we relax its budget to 400 queries, corresponding to 10 trajectories. All results are averaged over five repeated runs for each experimental configuration. We conduct experiments on a server with an AMD Ryzen 9 7950X 16-core processor, two NVIDIA GeForce RTX 4090 GPUs, and 64GB DDR4 RAM.

**Metrics.** We use *Matching Rate* (MR) (Cao et al., 2021; Xu et al., 2026) to evaluate the effectiveness of ownership verification, defined as the ratio of inputs correctly identified as the target fingerprints/watermarks. Also known as *Watermark (Success) Rate* (Cong et al., 2022; Lv et al., 2024b), matching rate reflects the similarity between the victim and suspect models. High uniqueness requires high matching rate on positive models and low matching rate on negative ones. Matching rate also measures fingerprint retention under attacks. For SSL-WM, we normalize its *Median Absolute Deviation* (MAD) metric to $[0, 1]$ to facilitate comparison with matching rate. Furthermore, we employ the ROC curve and AUC score to evaluate performance across varying thresholds. Regarding model utility, we report the *Clean Data Accuracy* (CDA) on downstream tasks to assess the performance on clean samples.

# E. Additional Experiment Results

## E.1. Stealthiness

*Table 4.* Detection rates of three methods on query samples on the CIFAR-100 dataset under the FTLL setting.

| Training Method | Detection Method | SSL-WM | MEA | ADV-TRA | MFUE | **Ours** |
|---|---|---|---|---|---|---|
| SimCLR | SHE | 0.46 | 0.58 | 0.79 | 0.15 | 0.32 |
| | TeCo | 0.77 | 0.79 | 0.18 | 0.03 | 0.09 |
| | ANP | 0.70 | 0.65 | 0.34 | 0.00 | 0.17 |
| MoCoV2 | SHE | 0.40 | 0.46 | 0.72 | 0.19 | 0.26 |
| | TeCo | 0.73 | 0.82 | 0.21 | 0.01 | 0.13 |
| | ANP | 0.75 | 0.69 | 0.39 | 0.02 | 0.14 |

Attackers in the deployment phase may deploy anomaly detection mechanisms to filter query samples, thereby attempting to evade IP verification. Consequently, ensuring the **stealthiness** of query samples is a critical prerequisite for the practicality of verification schemes. To evaluate this property, we assess the anomaly detection rates of our generated samples against three representative detection methods: SHE (Zhang et al., 2023), TeCo (Liu et al., 2023), and ANP (Wu & Wang, 2021).

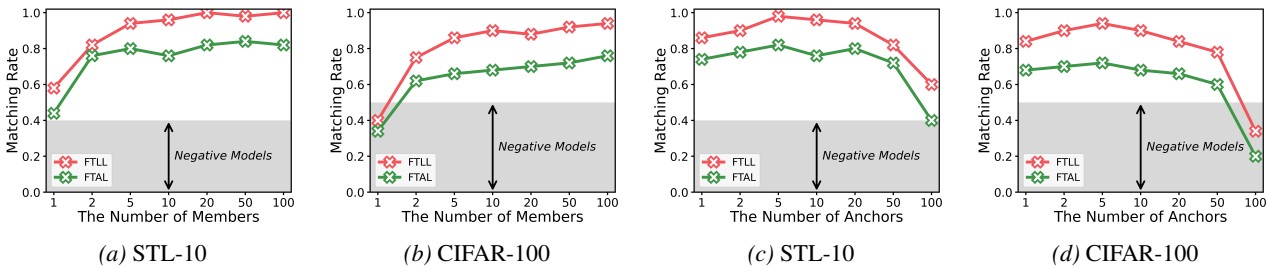

|  |  |  |  |
|---|---|---|---|
| *(a)* STL-10 | *(b)* CIFAR-100 | *(c)* STL-10 | *(d)* CIFAR-100 |

*Figure 9.* Matching rates while varying (a,b) the number of members in a group and (c,d) the number of anchors. The gray area represents the distribution range of matching rates for negative models.

SHE achieves OOD detection via a "store-then-compare" paradigm that measures the Modern Hopfield Energy discrepancy between test sample embeddings and stored in-distribution prototypes. We adopt the bottom $10\%$ quantile of the energy scores from normal samples as the threshold to identify anomalies. TeCo detects backdoor samples in a hard-label black-box setting by exploiting the anomalous inconsistency of model robustness across diverse image corruptions. ANP sanitizes backdoored models by identifying and pruning neurons exhibiting anomalous sensitivity to adversarial perturbations using limited clean data. We set both the trade-off coefficient $\alpha$ and the pruning threshold to $0.2$, following the recommended configuration for identifying backdoor-related neurons

Table 4 presents the detection rates of five methods on the CIFAR-100. Our method achieves superior stealthiness comparable to the state-of-the-art MFUE. SSL-WM and MEA fail against backdoor checks. They exhibit high detection rates on TeCo ($> 0.70$) and ANP ($> 0.65$), indicating that their fixed patterns fail to maintain robustness consistency and strongly activate sensitive neurons associated with backdoors. Furthermore, we find that TeCo, as an effective tool for identifying backdoor samples, successfully detects backdoor-style watermarks like SSL-WM and MEA, yet is nearly incapable of detecting fingerprint samples. This suggests that fingerprinting methods (ADV-TRA, MFUE, and our method) generally possess superior stealthiness across the majority of detection paradigms. Among them, MFUE proves to be the least identifiable, primarily because it employs samples characterized by non-adversarial perturbations. By maintaining consistently low detection rates across SHE ($0.26 \sim 0.32$), TeCo ($0.09 \sim 0.13$), and ANP ($0.14 \sim 0.17$) , our method demonstrates that it preserves natural robustness consistency and feature distribution patterns without triggering hypersensitive neurons, confirming its practicality for covert verification.

### E.2. Ablation Studies

In this section, we conduct a comprehensive ablation study to evaluate the impact of key hyperparameters of our fingerprinting scheme. Specifically, we investigate the sensitivity of verification performance to four critical factors: the group size, the number of anchors used for adversarial shifting, the complexity of downstream tasks (number of classes), the decision threshold for voting, the perturbation budget, and the number of clusters . Unless otherwise stated, these analyses are performed on the STL-10 and CIFAR-100 datasets to identify optimal configurations.

#### E.2.1. IMPACT OF THE GROUP SIZE.

We first examine how the group size affects the reliability of our group voting mechanism. Figure 9a and 9b illustrate the verification performance as the number of members increases from 1 to 100. When the group size is small (e.g., 1 member), the matching rate hovers around $0.4 \sim 0.6$, often falling into the negative models range (grey area). This confirms that single-point fingerprinting is susceptible to randomness and decision boundary shifts. However, as the number of members increases to 10, the matching rate surges to over 0.9 (FTLL) and stabilizes. This validates the effectiveness of our voting strategy: by aggregating predictions from a cluster of samples, we mitigate the stochasticity of individual outliers. We select 10 as the default setting, achieving a sweet spot between high verification confidence and low query budget.

#### E.2.2. IMPACT OF THE NUMBER OF ANCHORS.

Figure 9c and 9d report the performance with varying numbers of anchors used to guide the adversarial shifting. It can be seen that the matching rate initially improves as the number of anchors increases, peaking at approximately 10 anchors (reaching near 1.0 under FTLL). However, a notable performance drop occurs when the number of anchors becomes

*Table 5.* Matching rates with different $\epsilon$ values on STL-10 and CIFAR-100 datasets.

| Dataset | Strategy | 2/255 | 4/255 | 8/255 | 16/255 | 32/255 | 64/255 |
|---------|----------|-------|-------|-------|--------|--------|--------|
| STL-10 | FTLL | 0.12 | 0.26 | 0.86 | 1.00 | 1.00 | 1.00 |
| STL-10 | FTAL | 0.08 | 0.28 | 0.74 | 0.84 | 0.88 | 0.90 |
| CIFAR-100 | FTLL | 0.06 | 0.30 | 0.92 | 0.96 | 1.00 | 1.00 |
| CIFAR-100 | FTAL | 0.04 | 0.24 | 0.70 | 0.78 | 0.86 | 0.84 |

excessive (e.g., $> 50$), especially under the FTAL setting. This suggests that a moderate number of anchors is sufficient to define a stable target semantic region. Conversely, too many anchors may disperse the optimization objective, preventing the fingerprint samples from aggregating into a compact "feature island." Consequently, the loose cluster fails to maintain consistency during the drastic parameter updates of full fine-tuning. Thus, we set the anchor number to 10 to ensure precise feature positioning.

### E.2.3. IMPACT OF THE NUMBER OF CLASSES.

To assess the scalability of our method across diverse downstream scenarios, we evaluate verification performance on CIFAR-100 subsets with varying class counts ($N \in \{2, 5, \ldots, 100\}$), as shown in Figure 10a. Under the FTLL setting, our method exhibits high stability, maintaining a matching rate $> 0.9$ regardless of $N$. However, under the more aggressive FTAL setting, the matching rate declines as the number of classes increases (e.g., dropping to $\sim 0.68$ at $N = 100$). Simultaneously, the matching rate of negative models (grey area) rises, narrowing the decision margin. This may be because fine-tuning on more classes induces more extensive parameter updates,

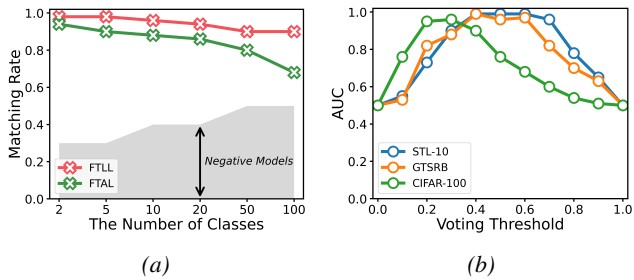

*(a)*            *(b)*

*Figure 10.* (a) The impact of the number of downstream classes on matching rates (CIFAR-100). (b) The sensitivity of verification performance (AUC) to the voting threshold across three datasets.

distorting the encoder's feature space and disturbing the fingerprint clusters. Additionally, a larger output space increases the probability of innocent models randomly mapping inputs to the target class. Nevertheless, for tasks with $N \leq 50$, our method maintains a distinct margin, covering most practical downstream scenarios.

### E.2.4. IMPACT OF THE VOTING THRESHOLD.

We investigate the sensitivity of the voting threshold $\tau$ used in our group voting mechanism. As illustrated in Figure 10b, the AUC scores across three datasets exhibit a distinct inverted U-shaped trend as $\tau$ varies from 0 to 1. Performance peaks and stabilizes for an interval but degrades significantly toward the extremes (approaching random guessing, i.e., AUC $\approx 0.5$, at $\tau = 0$ or $\tau = 1$). We observe that the optimal threshold tends to be lower for more complex datasets. This is likely because the expansive label space of CIFAR-100 increases the difficulty for members to achieve a unanimous consensus on a single class. This underscores the necessity of a balanced threshold. An overly low threshold ($\tau \approx 0$) accepts too many negative models (high false positive rate), while an overly strict threshold ($\tau \approx 1$) rejects valid but slightly perturbed positive models (high false negative rate).

### E.2.5. IMPACT OF THE PERTURBATION BUDGET.

To evaluate the sensitivity of our proposed Adversarial Shifting mechanism to the perturbation budget, we conduct an ablation study varying $\epsilon$ from 2/255 to 64/255. Table 5 presents the matching rate on the STL-10 and CIFAR-100 datasets under both FTLL and FTAL downstream adaptation strategies. As outlined in our methodology, $\epsilon$ limits the maximum allowable distortion during the group fingerprint generation. The empirical results demonstrate that an overly constrained budget (e.g., $\epsilon \leq 4/255$) is insufficient for successful ownership verification. Under these limited conditions, the optimization process lacks the requisite capacity to migrate base samples across the high-dimensional latent space. Consequently, the samples fail to tightly aggregate into a cohesive feature island, resulting in significantly low MRs (e.g., 0.24 on CIFAR-100 under FTAL for $\epsilon = 4/255$). Conversely, as $\epsilon$ is increased to 8/255 and our default configuration of 16/255, we observe a dramatic enhancement in verification performance. At $\epsilon = 16/255$, the fingerprinting method secures near-perfect MR under FTLL and demonstrates robust resilience (MR $> 0.78$) against the severe manifold distortions induced by FTAL.

*Table 6.* Matching rates with different $K$ values on STL-10 and CIFAR-100 datasets.

| Dataset | Strategy | 2 | 4 | 8 | 16 | 32 | 64 | 128 |
|---------|----------|------|------|------|------|------|------|------|
| STL-10 | FTLL | 0.72 | 0.98 | 1.00 | 1.00 | 0.98 | 1.00 | 0.98 |
| STL-10 | FTAL | 0.70 | 0.80 | 0.84 | 0.86 | 0.88 | 0.88 | 0.86 |
| CIFAR-100 | FTLL | 0.80 | 0.92 | 0.96 | 0.94 | 1.00 | 0.98 | 1.00 |
| CIFAR-100 | FTAL | 0.68 | 0.74 | 0.78 | 0.82 | 0.80 | 0.82 | 0.84 |

While further relaxing the budget beyond $32/255$ yields minor numerical gains, it inevitably compromises the visual stealthiness of the samples, risking exposure to query-side anomaly detection. Therefore, $\epsilon = 16/255$ establishes an optimal trade-off, providing enough adversarial capacity to construct immutable feature clusters while maintaining strict perceptual imperceptibility for the black-box setting.

E.2.6. IMPACT OF THE NUMBER OF CLUSTERS.

We further investigate the impact of the number of clusters $K$ utilized during the feature space characterization phase. By default, our implementation adopts $K = 8$, aligning with standard spectral clustering heuristics. Table 6 reports the verification performance across varying $K$ values, ranging from 2 to 128. Our empirical findings indicate that the granularity of the latent space partition significantly influences the robustness of the embedded fingerprint. When $K$ is strictly minimized (e.g., $K = 2$), the target clusters represent overly broad semantic regions. Anchors sampled from such dispersed areas lack the structural cohesion necessary to guide base samples into a tightly concentrated point, leading to suboptimal matching rates (e.g., $0.68$ on CIFAR-100 under FTAL). As we scale up $K$, the verification performance exhibits a consistent upward trend. This enhancement is especially pronounced for complex datasets with high semantic diversity, such as CIFAR-100, where the MR under the aggressive FTAL adaptation strategy rises from $0.78$ at $K = 8$ to $0.84$ at $K = 128$. The underlying mechanism for this improvement stems from refined manifold isolation. A larger $K$ forces the spectral clustering algorithm to partition the feature space into finer, more densely packed sub-manifolds. Selecting anchors from these highly localized regions ensures the construction of a substantially tighter feature island. This extreme intra-group proximity guarantees that all fingerprint members remain cohesively grouped and are deterministically assigned the same label by arbitrary downstream classification heads, thereby maximizing resistance to downstream fine-tuning.

# F. Observation

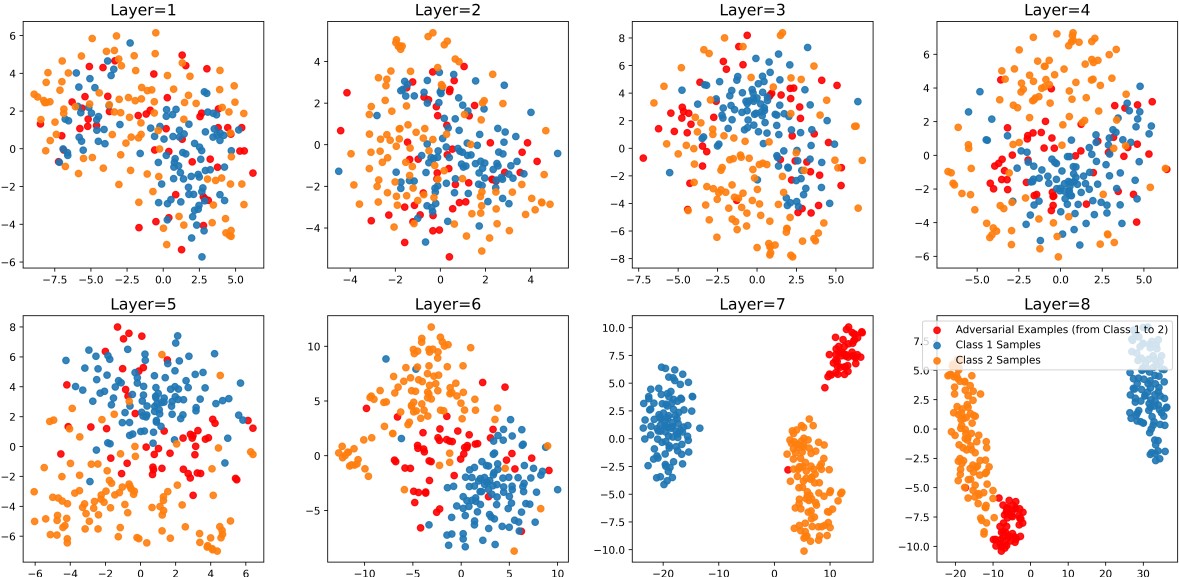

*Figure 11.* t-SNE visualization of feature representations across eight model layers. We visualize benign samples from two distinct classes alongside adversarial samples crafted to shift from Class 1 toward Class 2. As the depth increases, representations from different classes progressively segregate, eventually converging into well-defined clusters.

