# OpenReview forum: "Fingerprinting Pre-trained Encoders under Arbitrary Downstream Fine-Tuning via Adversarial Shifting"
_ICML.cc/2026/Conference — ICML 2026 regular_

### Official Review · Reviewer_4uaf · 2026-02-26

**Soundness:** 3
**Presentation:** 2
**Significance:** 3
**Originality:** 3
**Overall Recommendation:** 4
**Confidence:** 3

**Summary:**

This paper investigates the protection of intellectual property in pretrained models by constructing customized fingerprints. It targets a downstream-agnostic, black-box scenario, which more accurately reflects the practical pretraining–finetuning paradigm. The proposed method employs adversarial shifting to form stable fingerprint clusters in the latent space that remain robust after downstream finetuning. A group voting mechanism is further introduced to verify the fingerprints.

**Compliance With Llm Reviewing Policy:**

Affirmed.

**Final Justification:**

The authors' response addressed my concerns, and I'm willing to raise my score. Please incorporate the NLP verification pipeline and examples into the revised paper.

**Key Questions For Authors:**

Please see the weakness.

**Limitations:**

yes

**Strengths And Weaknesses:**

## Strength

1. The proposed downstream-agnostic, black-box setting is well-motivated and practical. I agree that in real-world scenarios, downstream data often have label sets distinct from those used in pretraining, and access to finetuned models is typically limited to APIs rather than full white-box access.

2. The problem setup presented in Section 2 is clear and easy to understand.

3. The idea of fingerprint clustering is empirically well-supported. Gathering samples of different classes towards a designated target anchor could be a strong fingerprint strategy.

## Weakness

1. Several components require more detailed ablation studies. For instance, it remains unclear why spectral clustering performs better than other distance-based methods, such as K-Means, if no results are provided.

2. The implementation details of the proposed method are somewhat vague and lack sufficient justification. For example, how is the density threshold determined? Are there any criteria for selecting the base samples? How are the verification thresholds set across different datasets?

3. The proposed framework involves numerous hyperparameters, making it difficult to optimize in practice. Some hyperparameters, such as the verification threshold, appear to vary significantly across datasets, which may limit the method’s generalizability.

4. The applicability of the proposed approach to natural language processing tasks (e.g., SNLI and MRPC) is unclear. Specifically, how is the PGD-optimized perturbation incorporated into textual data?

5. The results of CDA for various fingerprinting schemes (ADV-TRA, MFUE, and the proposed method) are missing. Consequently, the claim that *“the proposed method achieves superior fidelity (line 286)”* lacks empirical support.

6. Several expressions in the paper are unclear. For instance, some abbreviations (e.g., FTLL, FTAL, and CDA) are used before being defined. Moreover, the variable *c* in Equation (4) appears to be undefined.

---

> ### Author Rebuttal · Authors · 2026-03-30
>
> We thank the reviewer for the constructive feedback. We address each of your questions in detail below, and we welcome any further discussions during the rebuttal phase.
>
> **Response to W1:** Theoretically, as discussed in Section 3.2.1, K-Means assumes spherical clusters, whereas spectral clustering better captures the complex global connectivity inherent in self-supervised representation manifolds.
>
> Empirically, our preliminary experiments (supported by our submitted source code) confirm spectral clustering's superiority over K-Means, MiniBatch K-Means, and Gaussian Mixture Models (GMM), as shown below:
> |Clustering Method|Training Strategy|K-Means|MiniBatch K-Means|Spectral Clustering|GMM|
> |--|--|--|--|--|--|
> |STL-10|FTLL|0.90|0.86|1.00|0.44|
> |STL-10|FTAL|0.82|0.78|0.84|0.36|
> |CIFAR-100|FTLL|0.86|0.94|0.96|0.40|
> |CIFAR-100|FTAL|0.72|0.80|0.78|0.28|
>
> As shown, Spectral Clustering consistently achieves the highest and most stable matching rates across settings, empirically justifying our design choice. Furthermore, we have supplemented extensive ablations on cross-domain auxiliary datasets (see our response to Reviewer NmG7 Q1), the number of clusters $K$, and the perturbation budget $\epsilon$ (see our response to Reviewer 2kw1 Q4). All detailed results will be included in the revised appendix.
>
> **Response to W2:** We thank the reviewer for pointing this out. While these settings were previously distributed across sections, we will consolidate them in the revision to improve clarity. As stated in Appendix E, we retain clusters containing more than 50 samples. This explicitly filters out fragile outlier clusters (representing ~1% of the data) to minimize randomness.
>
> We randomly select base samples from non-target clusters ($\mathcal{C}_k \neq \mathcal{C}_{tgt}$). Because our adversarial shifting forcibly aggregates these samples into the dense target region, their initial feature-space positions have a negligible impact on the final intra-group consistency.
>
> As detailed in Appendix E, the voting threshold $\gamma_{vot}$ is set to 0.5 for STL-10/GTSRB and 0.3 for CIFAR-100. These values are empirically calibrated via ablation analysis (Appendix F.2.4) to optimally balance false positive and false negative rates —a standard heuristic practice in watermarking literature.
>
> **Response to W3:** This is a question worth discussing. Actually, our method is robust to hyperparameters and avoids tedious tuning. As shown in Appendix F.2, performance stabilizes once group size $N$ and anchors $M$ reach $\approx 10$ ; these default values generalize well across diverse datasets. The variation in voting thresholds reflects downstream label space complexity rather than a methodology flaw. For instance, CIFAR-100’s expansive label space naturally reduces random collision probability compared to STL-10, requiring a different optimal consensus threshold. Balancing these thresholds to mitigate false positives and negatives is a standard necessity in fingerprinting and watermarking literature. Our detailed ablations serve as a practical guide for deployment. Please refer to our response to Reviewer UXaD (Q3) for further discussion.
>
> **Response to W4:** Please refer to our response to Reviewer UXaD's Q1.
>
> **Response to W5:** We thank the reviewer for raising this point, which allows us to clarify the crucial distinction between watermarking and fingerprinting. Watermarking methods (e.g., SSL-WM, MEA) actively modify the encoder's weights during training via additional loss terms. This intrusive modification inevitably degrades the model's Clean Data Accuracy (CDA).
>
> Conversely, fingerprinting schemes—including ADV-TRA, MFUE, and our method—are strictly post-hoc verification mechanisms. They extract unique behavioral patterns without altering a single parameter of the victim encoder. Since model weights remain absolutely unchanged, the CDA of the protected encoder on any downstream task is strictly identical to the original model. Regarding our claim of “superior fidelity,” we refer to the ability to achieve robust verification without any impact on model utility. We will revise the presentation to make this distinction more precise in the final version.
>
> **Response to W6:** We sincerely thank the reviewer for their careful reading and apologize for the confusion, which resulted from space compression during formatting.
> FTLL (Fine-Tuning Last Layer) and FTAL (Fine-Tuning All Layers) refer to downstream fine-tuning strategies, while CDA stands for Clean Data Accuracy. Although their full definitions were provided in Appendix E, we will briefly define them at their first appearance in Section 4 of the revised main text.
> Regarding the variable $c$ in Equation (4), it represents any discrete class label output by the downstream classifier (i.e., $c \in \mathcal{C}_{down}$). We will add an explicit clarification immediately below Equation (4): "where $c$ denotes the class labels in the downstream label space."

---

> > ### Author Rebuttal · Reviewer_4uaf · 2026-04-01
> >
> > Thank you for the detailed response, which has addressed most of my concerns. However, the following issues remain:
> > - Regarding Weaknesses 2 and 3, I appreciate the authors for pointing out Appendix F, which includes ablation studies on several hyperparameters. However, I did not find experimental results on the verification threshold (defined in Eq. 5), which is the central concern I raised in the weakness section: "*How are the verification thresholds set across different datasets?*" and "*The verification threshold appears to vary significantly across datasets, which may limit the method's generalizability.*".
> > - Regarding Weakness 4, I now understand that the perturbations are optimized on sentence embeddings. However, although the authors state that embedding-to-token reconstruction is a well-established mechanism in adversarial NLP and is orthogonal to the paper's primary contribution, the success of the proposed method on NLP tasks still heavily relies on a strong and effective reconstruction tool. I encourage the authors to provide necessary implementation details and concrete examples to illustrate the verification process on NLP tasks for clearer elaboration and better reproducibility.

---

> > > ### Author Response · Authors · 2026-04-04
> > >
> > > **Response to W2&3:**
> > > We appreciate the opportunity to clarify the setting of the verification threshold ($\gamma_{vrf}$). We address your concerns below:
> > > To comprehensively evaluate distinguishability between positive/negative models, we report the Matching Rate (MR). Using raw matching proportions—rather than threshold-dependent binary verification success—is the standard practice in model IP protection [1-3]. $\gamma_{vrf}$ is essentially a post-hoc binary decision boundary applied after computing the MR. Furthermore, because baseline methods yield vastly different MR distributions and variances, establishing a single, universal verification threshold to concurrently evaluate all methods is impractical. Comparing raw MRs directly ensures a fairer and more comprehensive assessment.
> > > **How to Set $\gamma_{vrf}$.** Practically, any value strictly within the gap between positive and negative MR distributions serves as a valid decision boundary. As shown in Fig. 10 (evaluating varying group sizes/anchors), our positive MRs consistently exceed the negative MR distribution (gray area). In real-world deployments, defenders can set $\gamma_{vrf}$ to the midpoint of expected MRs, or locally simulate downstream adaptations to optimize metrics like the F1-Score.
> > > **Generalizability & Margin.** Most importantly, the nature of our threshold dynamic does not limit generalizability; rather, it highlights it. Please refer to Fig. 4b and our evaluations across diverse downstream tasks in Table 1. Our method demonstrates a distinct separation with a substantial margin (positive $\approx 0.8$ vs. negative $\approx 0.25$). Unlike baseline methods that suffer from severe MR overlap and thus require brittle, dataset-specific threshold tuning, our large margin provides an exceptionally broad and flexible range for selecting $\gamma_{vrf}$. This implies that a conservatively chosen threshold could reliably generalize across diverse datasets without strict recalibration.
> > >
> > > **Response to W4:**
> > > While our original NLP evaluation used the continuous embedding space, we agree that practical black-box deployment requires discrete reconstruction. To bridge continuous adversarial shifting and discrete tokens, we employ a Nearest Neighbor Search ([NNS](https://github.com/QData/TextAttack)) [4]. We will add the following end-to-end verification pipeline to the revised manuscript, illustrated with a concrete example from the SNLI dataset:
> > > - Step 1: Continuous Shifting (Embedding Level): For a base text $z$, we optimize its adversarial target $x_{mem}$ in the victim encoder's embedding space ($E_{vct}$) via PGD (Eq. 2-3) to reside firmly within a cohesive "feature island".
> > >   - Example: Given the original text "A man in a blue shirt is standing in front of a building." (Original Prediction: Neutral), its embedding is PGD-shifted toward the target cluster to obtain $x_{mem}$.
> > > - Step 2: Discrete Reconstruction (Token Level): To map $x_{mem}$ to a valid sequence, we perform a constrained NNS. We iteratively replace tokens to find the word $w^{\ast}$ minimizing the $L_2$ distance to $x_{mem}$, formulated as $w^{\ast} = \arg\min_{w \in \mathcal{V}} \|x_{mem} – E_{vct}(\dots w \dots)\|_2$. We logically constrain the search vocabulary with part-of-speech checking and semantic similarity scoring to maintain syntactic fluency and fidelity.
> > >   - Example: The constrained NNS identifies valid replacements to semantically approximate the shift, generating the discrete fingerprint text $z_{mem}$: "A guy in a blue jacket is standing before a building."
> > > - Step 3: Verification (Classification Level): The reconstructed texts form a group $\mathcal{G}$ queried against the suspect model $f_{spc}$. Because their underlying features tightly aggregate in the victim's space, any downstream classifier maps them to identical decision regions, triggering a successful Group Vote.
> > >   - Example: The suspect model outputs the verification prediction "Entailment" for $z_{mem}$. This prediction identically aligns with the consensus label of the target feature island members, enabling us to calculate the MR via group voting.
> > >
> > > By integrating NNS, discrete sequences retaining continuous adversarial properties could be generated. Notably, while this simple NNS approach is highly effective, adopting more advanced embedding-to-text mapping techniques could further optimize this adversarial transfer while fully preserving readability and content.
> > >
> > > We hope these address your concern and warmly welcome any further discussion.
> > >
> > > **Reference**
> > > [1] MEA-Defender: A robust watermark against model extraction attack. IEEE S&P, 2024.
> > > [2] SSL-WM: A black-box watermarking approach for encoders pre-trained by self-supervised learning. NDSS, 2022.
> > > [3] United we stand, divided we fall: Fingerprinting deep neural networks via adversarial trajectories. NeurIPS, 2024.
> > > [4] Textattack: A framework for adversarial attacks, data augmentation, and adversarial training in nlp. EMNLP, 2020.

---

### Official Review · Reviewer_2kw1 · 2026-03-10

**Soundness:** 2
**Presentation:** 2
**Significance:** 3
**Originality:** 3
**Overall Recommendation:** 4
**Confidence:** 4

**Summary:**

This paper proposes a downstream-agnostic, label-only fingerprinting method for verifying ownership of pre-trained encoders under arbitrary downstream adaptation. The core idea is to adversarially shift a group of base samples in the input space so that their encoder representations collapse into a dense target “feature island” guided by unsupervised anchors. Because these members are tightly clustered in the encoder feature space, they tend to elicit consistent labels under arbitrary downstream heads, enabling black-box verification via a group voting statistic. Experiments across multiple pre-training paradigms (SimCLR, MoCoV2, supervised learning, and SigLIP) and tasks (CV and NLP) report high matching rates and robustness against fine-tuning, pruning, model extraction, and input-time perturbations, with minimal overhead and without access to embeddings or logits.

**Compliance With Llm Reviewing Policy:**

Affirmed.

**Final Justification:**

The paper proposes a novel and logically coherent fingerprinting framework that shifts verification from fragile label matching to latent feature-space consistency, effectively addressing the critical challenge of black-box ownership verification for stolen pre-trained encoders. Rigorously validated through extensive experiments across diverse paradigms and adversarial stress-tests, the method demonstrates strong empirical robustness and offers a highly practical, downstream-agnostic solution for real-world model protection.

**Key Questions For Authors:**

1. Since the fingerprint queries are adversarially shifted only with respect to the victim encoder, what evidence supports the claim that they do not transfer to independently trained innocent encoders as well? Please provide either theoretical intuition or targeted empirical evidence on this point.

2. How exactly is adversarial shifting instantiated for NLP encoders (BERT) given discrete inputs? If embedding-space shifts are used, how are they mapped back to valid token sequences while preserving semantics and stealthiness?

3. The update rule in Eq. (3) uses sign(∇) under an $\ell_2$ constraint. Can you clarify whether the implementation uses $\ell_{\infty}$-PGD or normalized gradients for $\ell_2$? Please provide the precise implementation details and explain their effect on fidelity and verification performance.

4. What are the default values of $L$ (number of groups), $K$ (clusters), and $\varepsilon$ in your implementation, and how sensitive are MR and AUC to these hyperparameters? Please include main-text ablations or a short summary of the key appendix findings.

**Limitations:**

No. The paper would benefit from a discussion of adaptive attacks that explicitly push samples away from the anchor region / feature island, as well as possible countermeasures against such attacks.

**Strengths And Weaknesses:**

## Strengths

1. Completeness and rigor of the evaluation protocol. The paper conducts extensive experiments, covering multiple pre-training paradigms ranging from SimCLR and MoCoV2 to SigLIP and spanning both vision and language settings, including image classification and natural language inference. Moreover, the authors stress-test the method under a variety of adversarial scenarios, including full-model fine-tuning (FTAL), aggressive pruning, and model extraction attacks, providing substantial empirical evidence for the persistence and robustness of the proposed fingerprint.

2. Consistency between problem formulation and methodological reasoning. The paper is logically well-organized. It clearly explains why traditional fingerprinting schemes fail under label-space shift and, from this observation, motivates the transition from explicit label matching to latent feature-space consistency as the key verification signal. The threat model is closely aligned with the methodological design, and the overall narrative remains focused and coherent.

3. The problem addressed by the paper is practically meaningful. Pre-trained encoders are valuable foundation-model components, and once they are stolen and reattached to arbitrary downstream heads, conventional fingerprinting methods that rely on fixed label semantics often break down. Therefore, studying a verification mechanism that remains applicable under black-box, label-only, and downstream-agnostic conditions is indeed a potentially impactful direction.

4. The originality of the paper mainly lies in its angle of attack. Instead of continuing along the traditional route of fixed-label matching or internal-feature access, the authors attempt to construct compact clusters in feature space and then leverage intra-group prediction consistency for black-box verification. Combining adversarial shifting with group voting in the context of ownership verification for pre-trained encoders is a novel and interesting idea.

## Weaknesses

1. Insufficient justification for low false-positive behavior on innocent models. The fingerprint queries are generated by adversarially shifting base samples so that they approach a target feature island in the victim encoder’s representation space. However, it remains unclear why these same queries would not also become concentrated—or induce similarly high label consistency—under independently trained innocent encoders. In particular, the paper does not sufficiently explain why an innocent encoder could not form a feature island similar to that of the victim model, which would allow the group fingerprints to remain effective even on unrelated models. While the empirical results suggest a low false-positive rate, the paper does not provide a clear theoretical argument or targeted experimental analysis explaining why optimization toward a feature island of the victim model should remain non-transferable to unrelated encoders. As a result, the mechanism underlying discriminability against innocent models is not yet fully substantiated.

2. The paper claims to optimize the fingerprint generation objective under an $\ell_2$-norm constraint, yet the PGD update operator in Equation (3) uses a sign gradient, which is typically associated with $\ell_{\infty}$-style optimization. In addition, the definition of the number of anchors is not fully consistent: the summation term in Equation (2), the description in Algorithm 1, and the stated value of $M$ are not fully aligned.

3. Lack of transparency in the NLP implementation path. Although the experimental section includes BERT and downstream NLP tasks, the paper does not explain how the proposed gradient-based adversarial shifting is actually carried out in discrete token space. For example, it is unclear whether the method uses synonym substitution, gradient-guided token replacement, optimization in embedding space followed by input reconstruction, or some other mechanism. This missing detail substantially weakens the clarity and reproducibility of the NLP component.

4. Limited clarity and analysis regarding key hyperparameters. The paper does not clearly present the default values of several important hyperparameters, including the number of fingerprint groups $L$, the number of clusters $K$, and the perturbation budget $\varepsilon$. More importantly, although some sensitivity analyses appear to be included in the appendix, the main paper does not provide a sufficiently clear summary of the sensitivity of MR and AUC to these three hyperparameters. Since the effectiveness of the method may depend on these design decisions, the current presentation leaves its robustness and reproducibility insufficiently supported.

---

> ### Author Rebuttal · Authors · 2026-03-30
>
> We thank the reviewer for the constructive feedback, which are valuable for refining our work. We address each of your questions in detail below, and welcome any further discussions during the rebuttal phase.
>
> **Response to Q1&W1:** Thanks for this question about the uniqueness of our fingerprints. From the perspective of decision boundaries, while adversarial examples do exhibit some transferability, this primarily manifests in regions where different models share similar boundaries. However, even when two models share the exact same architecture and training data, inherent training randomness ensures their decision boundaries are never completely identical [1,2]. This is exactly why existing fingerprinting/watermarking methods can leverage adversarial examples near decision boundaries to mark these unique topological differences, thereby achieving IP verification. By shifting the fingerprint samples, they are forced to aggregate within the victim encoder's feature space. To an independent innocent encoder, these tailored perturbations largely act as random noise, causing the grouped samples to scatter rather than cluster in its latent space. This inherently low transferability of targeted adversarial perturbations is precisely why they are widely adopted to uniquely fingerprint models.
>
> Furthermore, our empirical results also comfirm this on negative models (i.e., independently trained encoders), which consistently yield near-random matching rates and low AUC (Fig. 4), in sharp contrast to victim-derived models.
>
> **Response to Q2&W3:** We thank the reviewer for raising this question. Following the operational scenario established by SSL-WM [3] and other works [4,5], our evaluation assumes that the model owner can directly access the suspect model's embedding layer during the verification phase. Please also refer to our detailed response to Reviewer UXaD's Q1. In the revised manuscript, we will explicitly detail this embedding-level operating mechanism in the Implementation Details and candidly discuss the lack of discrete token mapping as a limitation.
>
> **Response to Q3&W2:** Thanks for the reminder. This is a typo on our part, and we will correct it in the revised manuscript. Please refer to our response to Reviewer UXaD, Q2, which includes the exact code snippet for the L2 projection used in our experiments.
>
> **Response to Q4:** Thanks for this question. In our implementation, the default values are $L=10$ and $\epsilon=16/255$ (see Appendix E), while the cluster count $K=8$ follows the default parameter of the spectral clustering algorithm.
> Since the final MR is calculated by averaging the verification results across $L$ groups, increasing $L$ mainly serves to reduce variance rather than boost the absolute accuracy. Our experiments show that $L=10$ is sufficient to provide highly stable statistical significance.
>
> The results of our new ablation studies on perturbation budget $\epsilon$ are shown below:
> |$\epsilon$|Training Strategy|2/255|4/255|8/255|16/255|32/255|64/255|
> |--|--|--|--|--|--|--|--|
> |STL-10|FTLL|0.12|0.26|0.86|1.00|1.00|1.00|
> |STL-10|FTAL|0.08|0.28|0.74|0.84|0.88|0.90|
> |CIFAR-100|FTLL|0.06|0.30|0.92|0.96|1.00|1.00|
> |CIFAR-100|FTAL|0.04|0.24|0.70|0.78|0.86|0.84|
>
> When $\epsilon$ is too small (e.g., $\le 8/255$), adversarial shifting lacks the capacity to tightly aggregate members into a "feature island", leading to low MR (e.g., $\sim 0.24$ on CIFAR-100 under FTAL). Therefore, within the constraints of imperceptibility, a larger $\epsilon$ (e.g., $16/255$) ensures optimal and robust verification.
>
> We also conducted an ablation study on the number of clusters $K$, and the results are as follows:
> |K|Training Strategy|2|4|8|16|32|64|128|
> |--|--|--|--|--|--|--|--|--|
> |STL-10|FTLL|0.72|0.98|1.00|1.00|0.98|1.00|0.98|
> |STL-10|FTAL|0.70|0.80|0.84|0.86|0.88|0.88|0.86|
> |CIFAR-100|FTLL|0.80|0.92|0.96|0.94|1.00|0.98|1.00|
> |CIFAR-100|FTAL|0.68|0.74|0.78|0.82|0.80|0.82|0.84|
>
> It can be seen that appropriately increasing $K$ further improves MR, particularly for complex datasets like CIFAR-100 (MR under FTAL increases from $0.78$ at $K=8$ to $0.84$ at $K=128$). This is because a larger $K$ partitions the latent space into finer, more cohesive semantic clusters, providing tighter anchor regions that are more resilient to downstream fine-tuning.
>
> **Limitations:** Please refer to our response to NmG7 regarding the limitations point and Q2.
>
> **Reference**
> [1] Can Neural Nets Learn The Same Model Twice? Investigating Reproducibility and Double Descent From The Decision Boundary Perspective. CVPR, 2022.
> [2] Understanding Deep Learning Via Decision Boundary. IEEE TNNLS, 2023.
> [3] SSL-WM: A Black-Box Watermarking Approach For Encoders Pre-Trained By Self-Supervised Learning. NDSS, 2024.
> [4] Soft Prompt Threats: Attacking Safety Alignment and Unlearning in Open-Source LLMs Through The Embedding Space. NeurIPS, 2024.
> [5] Efficient Adversarial Training in LLMs with Continuous Attacks. NeurIPS, 2024.

---

> > ### Author Rebuttal · Reviewer_2kw1 · 2026-04-03
> >
> > The authors have addressed my concerns

---

> > > ### Author Response · Authors · 2026-04-04
> > >
> > > We sincerely thank you for your constructive feedback and for confirming that your concerns are fully resolved. We will carefully incorporate these refinements into the final manuscript. Given that all issues have been adequately addressed, we would be deeply grateful if you might consider reflecting this resolution in your overall assessment. Thank you again for your time!

---

### Official Review · Reviewer_UXaD · 2026-03-11

**Soundness:** 3
**Presentation:** 4
**Significance:** 3
**Originality:** 3
**Overall Recommendation:** 4
**Confidence:** 3

**Summary:**

The paper proposes a downstream-agnostic, label-only fingerprinting scheme for pre-trained encoders, aiming to verify ownership even after arbitrary downstream fine-tuning. The core idea is to generate group fingerprints by adversarially shifting a set of base samples in the encoder’s feature space toward dense anchor regions, so that any downstream head will map the group to the same output label with high consistency. Ownership is validated via a black-box group-voting mechanism that checks the concentration of predicted labels across multiple disjoint groups. Extensive experiments across CV and NLP settings indicate strong robustness to fine-tuning, pruning, input transformations, and some model extraction attacks, with low overhead and without modifying the pretraining process.

**Compliance With Llm Reviewing Policy:**

Affirmed.

**Final Justification:**

The authors have addressed my concerns and regarding their promises to update the manuscript I maintain my current score.

**Key Questions For Authors:**

1. How exactly is adversarial shifting instantiated for NLP? Please detail the perturbation space, constraints for imperceptibility, and the optimization procedure; provide examples and detection/stealth analyses.
2. The shifting objective uses an L2 constraint but a sign gradient step. Did you experiment with L2-normalized PGD steps, and how does this affect success, perceptual quality, and runtime? Also, should the loss target a centroid or nearest anchor rather than summing over anchors?
3. How sensitive are results to D_aux choice, anchor density threshold γ_ach, number of anchors M, group size N, and ε? Please include a main-text summary and guidelines for setting these hyperparameters.
4. What is the cross-domain robustness when the attacker’s downstream data are from a domain far from D_aux (e.g., medical images vs. natural images)? Could anchors learned from D_aux become unreliable, and can multi-domain anchors help?

**Limitations:**

yes

**Strengths And Weaknesses:**

Strengths:
1. Leverages adversarial shifting to create tightly clustered, endogenous fingerprints in feature space, decoupling verification from specific downstream labels or heads.
2. Introduces a simple yet effective label-only group voting protocol that relies on intra-group output consistency rather than logits or internal embeddings.
3. Anchor selection via unsupervised clustering to target dense, presumably stable regions of the representation manifold is a sound heuristic that connects to manifold stability under fine-tuning.
4. The high-level threat model and the generation/verification pipeline are clearly illustrated, and the intuition behind feature islands and downstream consistency is well motivated.
5. The verification protocol is straightforward and likely easy to reproduce.

Weaknesses:
1. The adversarial shifting objective and optimization details have inconsistencies as the L2 constraint is paired with a sign step (more typical for L∞), and the loss sums distances to all anchors per sample, which is unusual and may not match the intended centroid/nearest-anchor target.
2. NLP pipeline details are underspecified. Generating imperceptible “perturbations” for text to achieve feature-space shifting is non-trivial. The paper reports NLP results but does not explain how shifting is performed or how perceptual constraints are handled.
3. Adaptive attacker strategies tailored to the proposed protocol are not thoroughly explored.
4. The auxiliary dataset assumption and cross-domain generalization are not stress-tested under strong domain shift.

---

> ### Author Rebuttal · Authors · 2026-03-30
>
> We sincerely thank the reviewer for the constructive feedback, which are highly valuable for refining our work. We address each of your questions in detail below, and we welcome any further discussions during the rebuttal phase.
>
> **Response to Q1:** In our NLP setting, we follow the attack setup of SSL-WM [1] and works [2, 3], where the attack is performed in the continuous embedding space of the encoder. Specifically, we optimize perturbations on sentence embeddings via projected gradient descent (Eq. 2-3), bounded by an L2-norm constraint to ensure minimal representational deviation.
>
> To map these continuous vectors back to valid, stealthy discrete token sequences, practical solutions exist in recent NLP adversarial literature [4-6]. These methods successfully project embedding perturbations back to discrete tokens via nearest-neighbor search or constrained decoding, often utilizing language-model-based filtering to preserve semantic fluency. We emphasize that this embedding-to-token reconstruction is a well-established mechanism in adversarial NLP. It is strictly orthogonal to our paper's primary contribution, which focuses on designing a robust, downstream-agnostic IP verification framework within the feature space.
>
> **Response to Q2&W1:** We thank the reviewer for the careful reading. The $\text{sign}(\cdot)$ in Eq. (3)  is indeed a typographical error. In our actual implementation, we consistently utilize $l_2$-normalized PGD steps with an $l_2$-ball projection, rather than $l_\infty$-style updates:
>
> ```
> def project_onto_l2_ball(delta, eps):
>     # flatten
>     orig_shape = delta.shape
>     delta_flat = delta.view(-1)
>     norm = torch.norm(delta_flat, p=2)
>     if norm <= eps or norm == 0:
>         return delta
>     else:
>         return (delta_flat * (eps / norm)).view(orig_shape)
> ```
>
> Therefore, the reported results in the paper (including success rate, perceptual quality, and runtime) already correspond to this L2-PGD setting. We will correct this notation in the revised manuscript.
>
> Regarding the loss target, we intentionally avoid optimizing fingerprint samples toward a single centroid or nearest anchor. Such designs would cause the fingerprints to collapse into a single point, making them effectively indistinguishable from the anchor itself. If all samples converge to the exact same location, the fingerprint degenerates into a fixed representation, losing diversity and becoming far less robust to downstream fine-tuning. This is empirically supported by our ablation study (Appendix F.2.2).
>
> In contrast, our multi-anchor objective relies on the collective structure and relative relationships among multiple samples. It encourages them to align with the overall distribution of a feature cluster, successfully forming a cohesive yet non-degenerate "feature island."
>
> **Response to Q3:** We have already provided detailed ablation studies regarding the group size $N$ (Appendix F.2.1) and the number of anchors $M$ (Appendix F.2.2). As demonstrated, performance stabilizes and achieves high confidence when both $N$ and $M$ reach approximately 10.
>
> For the auxiliary dataset ($D_{aux}$), our method demonstrates strong cross-domain robustness (see our response to Q4). The perturbation budget $\epsilon$ controls the standard trade-off between stealthiness and matching rate; our supplementary ablations (see response to Reviewer 2kw1 Q4) show optimal performance is achieved when $\epsilon \ge 8/255$. The anchor density threshold $\gamma_{ach}$ is simply used to filter out outlier clusters (e.g., discarding clusters with <50 samples, ~1% of data) to reduce variance. Following your suggestion, we will consolidate these empirical findings into a dedicated summary in the revised main text to facilitate easy parameter selection.
>
> **Response to Q4:** To evaluate cross-domain robustness, we conducted additional experiments using a medical dataset (BloodMNIST) to simulate a severe domain shift. Results demonstrate that anchors learned from $D_{aux}$ remain highly reliable. Since the pre-trained encoder captures generalized, domain-agnostic features, the constructed "feature islands" persist effectively even across vastly different downstream domains. For more details on cross-domain experiments, please refer to our response to Reviewer NmG7's Q1.
>
> **Reference**
> [1] SSL-WM: A Black-Box Watermarking Approach For Encoders Pre-Trained By Self-Supervised Learning. NDSS, 2024.
> [2] Soft Prompt Threats: Attacking Safety Alignment and Unlearning in Open-Source LLMs Through The Embedding Space. NeurIPS, 2024.
> [3] Efficient Adversarial Training in LLMs With Continuous Attacks. NeurIPS, 2024.
> [4] Attacking Large Language Models with Projected Gradient Descent. ICML, 2024.
> [5] Sentence Embedding Leaks More Information Than You expect: Generative embedding inversion attack to recover the whole sentence. ACL, 2023.
> [6] Text Embeddings Reveal (almost) As Much As Text. EMNLP, 2023.

---

> > ### Author Rebuttal · Reviewer_UXaD · 2026-04-04
> >
> > I thank the authors for their response. Although the authors responded adequately on other weaknesses but their response on w2 is not adequate. Other reviewers also pointed their concerns on this matter which means regardless the authors claim of the perturbation method to be trivial, actually it is not. So I would encourage the authors to clarify adequately about w2.

---

> > > ### Author Response · Authors · 2026-04-06
> > >
> > > We are sorry we couldn't address this point in detail during the first round due to the strict character limits.
> > >
> > > **Response to W2: Reconstruction Details for NLP Tasks**
> > > While our original NLP evaluation was conducted within the continuous embedding space, we fully agree that practical black-box deployment necessitates robust discrete text reconstruction. To bridge this gap, we could follow established mechanisms from recent adversarial text literature [1-3] that project continuous embedding perturbations back to valid discrete tokens. Specifically, we employ a naive yet highly effective constrained Nearest Neighbor Search (NNS) that iteratively maps optimized continuous targets back to discrete tokens, applying part-of-speech and semantic similarity checks to guarantee perceptual fidelity. Querying the suspect model with these reconstructed text sequences effectively maintains the required feature-space aggregation, enabling our Group Vote mechanism to reliably verify ownership in real-world scenarios.
> > > For a complete, step-by-step breakdown and a concrete example of this NLP verification pipeline, please kindly refer to our second Response to W4 for Reviewer 4uaf.
> > >
> > > We hope this clarifies our approach. Please let us know if you have any other questions—we would be happy to discuss them further.
> > >
> > > **Reference**
> > > [1] Attacking Large Language Models with Projected Gradient Descent. ICML, 2024.
> > > [2] Sentence Embedding Leaks More Information Than You expect: Generative embedding inversion attack to recover the whole sentence. ACL, 2023.
> > > [3] Text Embeddings Reveal (almost) As Much As Text. EMNLP, 2023.

---

### Official Review · Reviewer_NmG7 · 2026-03-15

**Soundness:** 3
**Presentation:** 2
**Significance:** 3
**Originality:** 3
**Overall Recommendation:** 4
**Confidence:** 3

**Summary:**

This paper’s topic is the protection of intellectual property for pre-trained neural network encoders in the modern pre-training and downstream fine-tuning paradigm. The paper explores a central aspect of model ownership verification: designing fingerprinting mechanisms that remain effective when stolen encoders are fine-tuned for arbitrary downstream tasks and accessed only through label-only black-box APIs.

The paper proposes a fingerprinting framework based on Adversarial Shifting, where fingerprint samples are generated by perturbing base inputs so that their embeddings migrate toward a dense cluster in the encoder’s feature space. These samples form group fingerprints, which are expected to produce consistent predictions across arbitrary downstream heads. Ownership is then verified through a group voting mechanism, which measures prediction consistency among fingerprint samples when querying a suspect model.

Experiments are conducted across several encoder architectures and training paradigms (e.g., supervised learning, SimCLR, MoCoV2, and SigLIP), as well as multiple datasets in both vision and NLP domains. The authors show that the proposed approach maintains relatively high matching rates under downstream fine-tuning, pruning, model extraction, and inference-time perturbations.

**Compliance With Llm Reviewing Policy:**

Affirmed.

**Key Questions For Authors:**

How sensitive is the method to the choice, size, and domain of the auxiliary dataset used for feature-space clustering?

How would the method perform against an adaptive attacker that explicitly tries to break the consistency of the fingerprint group during downstream fine-tuning?

**Limitations:**

No, please add the limitations of the proposed approach.

**Strengths And Weaknesses:**

Strengths:
The paper addresses the increasingly important problem of protecting the intellectual property of pre-trained encoders, which have become valuable assets due to the high computational and data costs required to train them. The problem setting, downstream fine-tuning with black-box access, is practical and realistic.

The key idea of shifting fingerprinting from label matching toward feature-space clustering and prediction consistency is conceptually interesting. This design allows verification without assuming knowledge of the downstream label space.

The proposed approach operates under a strict label-only black-box setting, which is stronger and more realistic than prior works that require access to logits or internal embeddings.

The experimental evaluation spans several model architectures, pre-training paradigms, and attack scenarios (fine-tuning, pruning, model extraction, and input transformations). The results generally demonstrate improved matching rates compared to existing baselines.

weaknesses:
The absence of a related work section in the main manuscript. While related work appears in the appendix, it is essential to include a concise but clear discussion of prior work in the main paper to properly contextualize the contribution. Without this, it is difficult to understand how the proposed method contrasts with existing fingerprinting and watermarking approaches.

The paper motivates the method by noting that adversarial perturbations exhibit small deviations in shallow layers but become amplified in deeper layers of the network. However, this phenomenon has been widely observed and discussed in prior adversarial robustness literature and should be properly referenced and contextualized.

The fingerprint generation process requires an auxiliary dataset to characterize the encoder’s feature space. The authors should sufficiently analyze the sensitivity of the approach to the domain, size, or distribution of this auxiliary dataset.

---

> ### Author Rebuttal · Authors · 2026-03-30
>
> We sincerely thank the reviewer for the constructive feedback, which are highly valuable for refining our work. We address each of your questions in detail below, and we welcome any further discussions during the rebuttal phase.
>
> **Response to W1:** We appreciate the reviewer's valuable feedback. Due to strict space limits, our related work is currently in Appendix B. In the revised main text, we will include a concise discussion to better contextualize our contributions. Specifically, we will explicitly contrast our downstream-agnostic, label-only approach with traditional label-dependent fingerprinting (e.g., ADV-TRA) and intrusive watermarking techniques (e.g., SSL-WM).
>
> **Response to W2:** We agree that layer-wise perturbation amplification is a well-studied phenomenon. In our revision, we will properly contextualize this by citing relevant adversarial robustness literature [1-3]. While prior works primarily analyze this phenomenon to understand or enhance attacks, our novelty lies in repurposing it for IP protection. Specifically, we leverage this deep-layer amplification to construct stable "feature islands" via adversarial shifting, effectively overcoming the downstream semantic shift that invalidates traditional label-based fingerprints.
>
> **Response to Q1:** Following suggestions from Reviewer NmG7 and UXaD, we assess cross-domain sensitivity using the BloodMNIST medical dataset [4] (17,092 medical images, 8 classes) across two scenarios (FTAL setting): (a) the defender trains and fingerprints on BloodMNIST, while the attacker fine-tunes on CIFAR-100; (b) the defender trains on GTSRB, fingerprints on BloodMNIST, and the attacker fine-tunes on CIFAR-100. The average matching rates of positive models under the FTAL setting are illustrated in the following [Figure in anonymous link](https://anonymous.4open.science/r/EncoderFingerprint-AF28/figs/fig_ab.png).
>
> As observed, while cross-domain setups slightly reduce matching rates across all methods, ours remains highly effective and consistently outperforms baselines. This resilience stems from the encoder's generalized representations, which preserve our "feature islands" despite downstream domain shifts.
> Regarding dataset size, performance drops slightly if the auxiliary dataset is too small (<1k samples) due to inadequate feature characterization, but quickly stabilizes at a moderate scale. Practically, defenders inherently possess their original training data (Scenario a), ensuring reliable feature-space shifting. Full results will be added to the revised appendix.
>
> **Response to Q2:** We thank the reviewer for this insightful question. To explicitly break the fingerprint group's consistency, an adaptive attacker would first need access to the fingerprint samples. However, in standard IP protection scenarios, the model owner or a trusted authority keeps these strictly confidential, preventing targeted removal.
>
> Furthermore, unlike traditional single-trigger fingerprints, our group-level approach forcibly embeds samples into dense, stable semantic clusters within the feature space. Disrupting this intrinsic consistency requires drastically altering the learned representations, which would inevitably degrade downstream model utility. Our experiments support this: under severe modifications—including full fine-tuning, high-ratio pruning (up to 80%), and embedding perturbations—our method maintains high matching rates, demonstrating strong robustness against aggressive adaptive alterations.
>
> **Limitations:** We thank the reviewer for this valuable suggestion. We will add a dedicated "Limitations and Discussion" section in the revision, including, but not limited to: (1) Extreme modifications: Highly destructive attacks (e.g., DFMS-HL or >80% pruning) can distort the feature space and degrade matching rates, though this concurrently destroys the model's inherent utility. (2) Large output spaces: For downstream tasks with numerous classes (e.g., >100), the verification margin narrows slightly, necessitating careful voting threshold calibration even as overall effectiveness remains intact. (3) NLP modality challenges: Mapping continuous adversarial shifts back to discrete token spaces while preserving semantics adds implementation complexity compared to continuous vision domains. (4) Potential adaptive attacks: Like prior post-hoc methods, we assume fingerprint secrecy. If these queries are exposed, attackers might attempt targeted removal, presenting a vital direction for future research.
>
> **Reference**
> [1] Defense Against Adversarial Attacks Using High-Level Representation Guided Denoiser. CVPR, 2018.
> [2] Democratic Training Against Universal Adversarial Perturbations. ICLR, 2025.
> [3] Pixel2Feature Attack (P2FA): Rethinking The Perturbed Space To Enhance Adversarial Transferability. ICML, 2025.
> [4] A Dataset of Microscopic Peripheral Blood Cell Images For Development of automatic Recognition Systems. Data in Brief, 2020.

---

> > ### Author Rebuttal · Reviewer_NmG7 · 2026-04-03
> >
> > The authors have addressed my concerns.

---

> > > ### Author Response · Authors · 2026-04-04
> > >
> > > We sincerely thank you for your constructive feedback and for confirming that your concerns are fully resolved. We will carefully incorporate these refinements into the final manuscript. Given that all issues have been adequately addressed, we would be deeply grateful if you might consider reflecting this resolution in your overall assessment. Thank you again for your time!

---

### Decision · Program_Chairs · 2026-04-30

**Decision:**

Accept (regular)

**Comment:**

4x weak accept. This paper proposes a downstream-agnostic, label-only fingerprinting framework for pre-trained encoders that uses adversarial shifting to construct feature-space fingerprint groups and verifies ownership through group voting after arbitrary downstream fine-tuning. The reviewers agree on the (1) practical and well-motivated black-box, downstream-agnostic problem setting, (2) novel and coherent shift from brittle label matching to feature-space consistency and group voting, and (3) broad empirical evaluation across multiple encoder paradigms, modalities, and attack scenarios showing strong robustness. However, they note (1) insufficient clarity and reproducibility in several implementation details, especially for the NLP pipeline and some hyperparameters, (2) limited justification or analysis for false-positive behavior, adaptive attacks, and threshold selection/generalizability, and (3) presentation issues such as missing or compressed related-work/contextual discussion, notation inconsistencies, and underexplained ablations. The authors’ rebuttal substantially addressed most concerns, with two reviewers explicitly marking them fully resolved, one reviewer indicating the response addressed the concerns and being willing to raise the score, and one reviewer still requesting clearer NLP reconstruction details despite otherwise positive feedback, so the AC leans to accept this submission.